# High-order Equivariant Flow Matching for Density Functional Theory Hamiltonian Prediction

**Seongsu Kim**[1]    **Nayoung Kim**[1]    **Dongwoo Kim**[2]    **Sungsoo Ahn**[1]

[1]KAIST    [2]POSTECH

{seongsu.kim, nayoungkim, sungsoo.ahn}@kaist.ac.kr

dongwoo.kim@postech.ac.kr

https://github.com/seongsukim-ml/QHFlow

## Abstract

Density functional theory (DFT) is a fundamental method for simulating quantum chemical properties, but it remains expensive due to the iterative self-consistent field (SCF) process required to solve the Kohn-Sham equations. Recently, deep learning methods are gaining attention as a way to bypass this step by directly predicting the Hamiltonian. However, they rely on deterministic regression and do not consider the highly structured nature of Hamiltonians. In this work, we propose QHFLOW, a high-order equivariant flow matching framework that generates Hamiltonian matrices conditioned on molecular geometry. Flow matching models continuous-time trajectories between simple priors and complex targets, learning the structured distributions over Hamiltonians instead of direct regression. To further incorporate symmetry, we use a neural architecture that predicts SE(3)-equivariant vector fields, improving accuracy and generalization across diverse geometries. To further enhance physical fidelity, we additionally introduce a fine-tuning scheme to align predicted orbital energies with the target. QHFLOW achieves state-of-the-art performance, reducing Hamiltonian error by 73% on MD17 and 53% on QH9 compared to the previous best model. Moreover, we further show that QHFLOW accelerates the DFT process without trading off the solution quality when initializing SCF iterations with the predicted Hamiltonian, significantly reducing the number of iterations and runtime.

## 1 Introduction

Density functional theory (DFT) [1, 2] is a cornerstone of modern computational physics [3, 4], chemistry [5, 6], and materials science [7, 8]. It offers a powerful trade-off between accuracy and computational efficiency in predicting the electronic properties of atoms, molecules, and solids [9–11]. The practical implementation of DFT relies mainly on solving the Kohn-Sham equations [12], which describe a system of non-interacting electrons that yield the same ground-state density as the interacting system [12]. These equations are typically solved using the self-consistent field (SCF) method [13–15], an iterative procedure that updates the electron density until convergence. To this end, at each iteration, the Kohn–Sham Hamiltonian must be reconstructed from the kinetic operator, external potential, Coulomb (Hartree) potential, and exchange–correlation potential [16, 17]. However, this reconstruction of the Hamiltonian becomes computationally demanding for larger systems, limiting the scalability of conventional DFT approaches [18].

To address this challenge, recent work has explored machine learning models to predict the Hamiltonian directly from atomic configurations, with the aim of bypassing or accelerating the SCF loop [19–26]. A key challenge in this task is ensuring that the model preserves the physical symmetries of molecular systems, particularly SE(3) symmetry, which governs how Hamiltonians transform

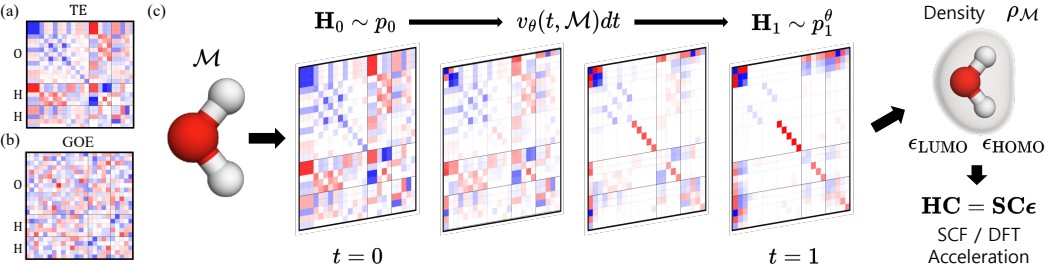

Figure 1: **Overview of QHFLOW.** (a) Initial Hamiltonian sampled from the tensor expansion-based SE(3)-invariant prior (TE). (b) Initial Hamiltonian sampled from the Gaussian orthogonal ensemble SE(3)-invariant prior (GOE). (c) QHFLOW transforms the initial Hamiltonian $\mathbf{H}_0$ into the target Hamiltonian $\mathbf{H}_1$ using flow matching, guided by an SE(3)-equivariant vector field $v_\theta(t, \mathcal{M})$ from an invariant prior $p_0$. The predicted Hamiltonian $\mathbf{H}_1$ defines a learned target distribution $p_1^\theta$ and is used to compute the electronic density $\rho_\mathcal{M}$, as well as the $\epsilon_{\text{LUMO}}$ and $\epsilon_{\text{HOMO}}$. When $\mathbf{H}_1$ is used to initialize SCF, QHFLOW can accelerate conventional DFT and SCF procedures.

under rotations and translations due to their construction in a spherical harmonics basis. SE(3)-equivariant networks such as PhisNet [20], QHNet [23], WANet [25], and SHNet [26] have been proposed to address this, using tensor representation and Clebsch–Gordan products to enforce rotational equivariance. These approaches rely on pointwise regression, which may lead to high predictive uncertainty and often struggle to capture the structural correlations present in the Hamiltonian matrix.

**Contribution.** In this work, we introduce QHFLOW, a high-order SE(3)-equivariant flow matching for generating Kohn-Sham Hamiltonians. Our work uses flow matching to learn a neural ordinary equation (ODE) that transforms a prior distribution to the target distribution of Hamiltonians. We also parameterize the vector fields using high-order SE(3)-equivariant neural networks to ensure that symmetry is preserved not only in the output, but throughout the entire ODE trajectory [27, 28].

To be specific, QHFLOW generates Hamiltonians conditioned on molecular geometries using equivariant architectures and is trained to match the distribution of Hamiltonians obtained from DFT. To this end, we design two types of SE(3)-invariant priors to support equivariant flow-based learning: a *Gaussian orthogonal ensemble (GOE)* and a *tensor expansion-based (TE)* prior that includes a group-theoretic structure. To further enhance the accuracy of energy-related properties, we introduce a fine-tuning stage that aligns the predicted orbital energies with those from the DFT, inspired by the weight alignment loss in WANet [25]. We provide an overview of QHFLOW in Figure 1.

Our main contributions are as follows:

- **Flow matching for Hamiltonian prediction**: We are the first to formulate Hamiltonian prediction as a generative problem, learning a trajectory to generate Kohn-Sham Hamiltonians conditioned on molecular geometry using SE(3)-equivariant vector fields.

- **Symmetry-aware prior distributions**: We introduce two SE(3)-invariant priors, based on the Gaussian orthogonal ensemble (GOE) and the tensor expansion (TE), ensuring equivariant flow matching and symmetrical initialization of Hamiltonian priors.

- **Energy alignment fine-tuning**: We introduce a fine-tuning strategy that aligns the predicted orbital energies with target orbital energies. This step encourages the model to match the target orbital energies, resulting in physically consistent property predictions.

- **Empirical performance**: Extensive experiments on MD17 and QH9 demonstrate that QHFLOW outperforms recent methods in terms of predicting the Hamiltonian MAE, orbital energy prediction, HOMO, LUMO and gap energy accuracy. We also show that using QH-FLOW's predictions as initialization for the SCF procedure leads to significant acceleration of the DFT process, reducing the number of iterations and total computation time.

## 2   Related work

**Hamiltonian matrix prediction.** Learning-based approaches for Hamiltonian matrix prediction have emerged as promising alternatives to the computationally demanding traditional DFT method.

Early work such as SchNOrb [19] extends SchNet [29] to enable accurate prediction of molecular orbitals. PhiSNet [20] introduces an architecture that explicitly respects SE(3)-symmetry using tensor representations and tensor product operations. QHNet [23] improves computational efficiency by reducing the number of high-order tensor products, and WANet [25] further extends this direction to larger molecular systems by incorporating a physically grounded loss function. Recently, SPHNet [26] improves scalability through adaptive path selection in the tensor product. Despite these advances, all preceding methods treat Hamiltonian prediction as a pointwise regression task. In contrast, our approach leverages flow matching to model the structural correlation in the Hamiltonian.

**Flow matching for prediction.** Flow matching [30, 31] has gained significant attention in generative modeling for its ability to leverage an ordinary differential equation (ODE)-based formulation, enabling faster inference than diffusion models [27]. Flow matching has been increasingly applied to machine learning problems in chemistry and drug discovery [32]. In particular, recent work has shown its effectiveness in regression-style applications such as protein structure prediction [33], crystal structure prediction [34, 35], and protein binding site identification [36], where modeling complex target distributions is crucial. In this work, we further extend flow matching to Hamiltonian matrix prediction. By framing the task as a distributional matching problem, our model learns structured outputs with greater accuracy and physical fidelity than conventional regression-based methods, illustrating the broader utility of flow matching in scientific prediction tasks.

## 3 Preliminary

### 3.1 Kohn-Sham density functional theory and Roothaan-Hall equation

**Kohn-Sham density functional theory (KS-DFT).** The Kohn-Sham (KS) equations [1, 12] are simplified analogs of the many-body Schrödinger equation [37] that retain a one-to-one correspondence with the ground-state density of the interacting electron system. This framework has revolutionized quantum mechanical simulations of electronic structures in atoms, molecules, and solids [9–11], offering a practical balance between computational efficiency and predictive accuracy.

Consider an $N$-particle system associated with the wavefunction $\Psi : \mathbb{R}^{3 \times N} \to \mathbb{C}$. In principle, the system is governed by the stationary Schrödinger equation $\hat{H}\Psi = \epsilon\Psi$, which describes the quantum behavior of a system using the Hamiltonian operator $\hat{H}$ and the total energy $\epsilon$. A direct solution is generally intractable, as its computational cost scales exponentially with the number of particles $N$.

Kohn-Sham density functional theory (KS-DFT) provides a computationally feasible alternative. Instead of solving the complex many-body wavefunction $\Psi$, KS-DFT uses the ground-state electron density $\rho(\mathbf{r})$ as its fundamental variable. This is achieved by introducing a fictitious system of non-interacting electrons that share the same ground-state density with the real, interacting wavefunction $\Psi$. The behavior of non-interacting system is described by a set of $N$ single-particle orbitals, $\{\psi_i\}_{i=1}^{N}$, which are the solutions to the KS equations:

$$\mathcal{H}[\rho]\psi_i = \epsilon_i\psi_i, \qquad \rho(\mathbf{r}) = \sum_{n=1}^{N} |\psi_i(\mathbf{r}_i)|^2, \tag{1}$$

where $\rho : \mathbb{R}^3 \to \mathbb{R}_+$ is the electron density, $\mathcal{H}[\rho]$ is the effective single-particle KS Hamiltonian constructed from $\rho$, and $\epsilon_i \in \mathbb{R}$ is the orbital energy. The $\mathcal{H}[\rho]$ contains the external, Hartree, and exchange-correlation terms. We provide a detailed explanation in Appendix A.1.

**Roothann-Hall (RH) equation.** In practice, the KS-equations are projected into a finite basis set, converting differential equations into the matrix eigenvalue problem known as RH-equation [38, 39]:

$$\mathbf{H}[\mathbf{C}]\mathbf{C} = \mathbf{S}\mathbf{C}\boldsymbol{\epsilon}. \tag{2}$$

Here, $\mathbf{C} \in \mathbb{R}^{B \times N}$ denotes the set of coefficients where each row $\{C_{bi}\}_{b=1}^{B}$ express an orbital $\psi_i$ as a linear combination of $B$ basis functions $\{\phi_b(\mathbf{r})\}_{b=1}^{B}$ *i.e.,* $\psi_i(\mathbf{r}) = \sum_{i=1}^{B} C_{bi}\phi_b(\mathbf{r})$. Next, $\mathbf{H}[\mathbf{C}]$ is the Hamiltonian matrix $\mathbf{H} \in \mathbb{R}^{B \times B}$ constructed by the coefficient $\mathbf{C}$, and $\boldsymbol{\epsilon} \in \mathbb{R}^{N \times N}$ is the diagonal matrix of orbital energies *i.e.,* $\mathtt{diag}(\epsilon_1, \ldots, \epsilon_N)$. Finally, $\mathbf{S} \in \mathbb{R}^{B \times B}$ is the matrix of overlap between basis functions $S_{bb'} = \int \phi_b^*(\mathbf{r})\phi_{b'}(\mathbf{r})d\mathbf{r}$ where $\phi_b^*$ denotes the complex conjugate of $\phi_b$. The basis functions $\{\phi_b(\mathbf{r})\}$ is selected to match the system geometry and commonly chosen between Slater-type orbitals (STOs) [40, 15], Gaussian-type orbitals (GTOs) [41], or plane waves [42, 10].

## 3.2 SO(3)-equivariance in RH-DFT

**SO(3)-equivariance and irreducible representations.** Formally, a map $f$ is said to be SO(3)-equivariant if $\mathbf{R} \cdot f(x) = f(\mathbf{R} \cdot x)$ for all $\mathbf{R} \in$ SO(3) where $\cdot$ denotes the SO(3) group action. This equivariance can be described in terms of the representation theory [43]. A *representation* of SO(3) is a matrix-valued function that describes the transforms under rotations, and it is called *irreducible* if it cannot be decomposed into smaller invariant subspaces. These *irreducible representations (irreps)* form the fundamental components for constructing SO(3)-equivariant function, and *irrep vector* is the vector that transforms under a specific irrep. Each SO(3) irrep is indexed by a non-negative integer $l$, and represented by a *Wigner D-matrix* $D^{(\ell)}(\mathbf{R}) \in \mathbb{C}^{(2\ell+1)\times(2\ell+1)}$. These matrices characterize how scalar ($\ell = 0$), vector ($\ell = 1$), and higher-order tensor features ($\ell \geq 2$) transform under rotation. A rank-$\ell$ irrep vector $\mathbf{w}^{(\ell)} \in \mathbb{R}^{(2\ell+1)}$ transforms under rotation $\mathbf{R}$ as: $\mathbf{w}^\ell \xrightarrow{\mathbf{R}} \mathbf{R} \cdot \mathbf{w}^{(\ell)} = D^{(\ell)}(\mathbf{R})\mathbf{w}^{(\ell)}$ where $\xrightarrow{\mathbf{R}}$ implies the transformation under $\mathbf{R}$. We provide additional background on group theory and equivariance in Appendices A.2 and A.3.

**High-order SO(3)-equivariance of RH-DFT.** The Hamiltonian matrix in RH-DFT exhibits a highly structured form of equivariance. As shown in Figure 2, the matrix can be organized into blocks, where each block corresponds to interactions between two sets of orbital basis functions indexed by pairs of quantum principal numbers and angular momentum numbers, i.e., $(n_1, \ell_1)$ and $(n_2, \ell_2)$. For example, a block labeled $1s$-$2p$ denotes interactions between $(n_1, \ell_1) = (1, 0)$ and $(n_2, \ell_2) = (2, 1)$. These orbital groupings induce a higher-order structure, where the Hamiltonian must remain equivariant not only at the global matrix level but also within each individual block. This makes Hamiltonian modeling more challenging than conventional equivariant tasks such as force or energy prediction, as it requires explicit handling of block-level symmetry constraints.

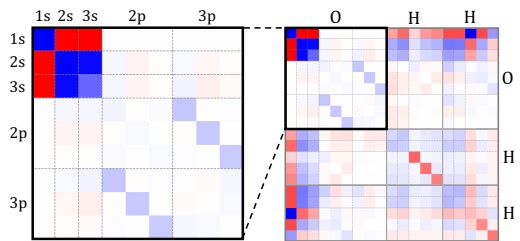

Figure 2: **Illustration of the Hamiltonian matrix structure of $H_2O$.** Rows and columns are indexed by quantum and angular momentum numbers (*i.e.*, $1s$, $2s$) ordered by atom (O, H, H). The full matrix (right) is partitioned into atomic blocks with gray dashed line; the top-left $9 \times 9$ sub-matrix corresponds to O and is shown in detail (left), grouped by quantum and angular momentum. Colors indicate the sign and magnitude of matrix elements.

This symmetry arises naturally in RH-DFT when the basis set is constructed from *spherical harmonics*, which serve as irrep vectors of SO(3). In particular, standard basis functions such as STO or GTO are defined using spherical harmonics $Y_\ell^m$, where $\ell$ and $m$ denote the angular momentum and magnetic quantum numbers, respectively [44]. When using such spherical-type basis, the full orbital basis set $\{\phi_b\}_{b=1}^B$ can be partitioned into $K$ groups, with the $k$-th group $\{\phi_{n_k\ell_k m}\}_{m=-\ell_k}^{\ell_k}$ sharing the same principal quantum number $n_k$ and angular momentum quantum number $\ell_k$. Each group spans a $(2\ell_k + 1)$ functions indexed by $m$, forming a rank-$\ell_k$ irrep vector. We assume the columns of the coefficient matrix $\mathbf{C}$ are ordered according to these groupings. Under a rotation $\mathbf{R} \in$ SO(3), the matrix $\mathbf{C}$ transforms as follows:

$$\mathbf{C} \xrightarrow{\mathbf{R}} \mathbf{R} \cdot \mathbf{C} = \mathcal{D}(\mathbf{R})\mathbf{C}, \qquad \mathcal{D}(\mathbf{R}) = D^{(\ell_1)}(\mathbf{R}) \oplus \cdots \oplus D^{(\ell_K)}(\mathbf{R}), \tag{3}$$

where each $D^{(\ell_k)} \in \mathbb{R}^{(2\ell_k+1)\times(2\ell_k+1)}$ is the rank-$\ell_k$ Wigner D-matrix and $\oplus$ denotes the direct sum over irreps. The Hamiltonian matrix $\mathbf{H}$, being a function of $\mathbf{C}$, also satisfies the SO(3)-equivariance:

$$\mathbf{H}\left[\mathbf{C}\right] \xrightarrow{\mathbf{R}} \mathbf{R} \cdot \mathbf{H}\left[\mathbf{C}\right] = \mathbf{H}\left[\mathcal{D}(\mathbf{R})\mathbf{C}\right] = \mathcal{D}(\mathbf{R})\mathbf{H}\left[\mathbf{C}\right]\mathcal{D}(\mathbf{R})^{-1}. \tag{4}$$

We provide further details about the SO(3)-equivariance structure of Hamiltonian submatrix and the direct sum of Wigner D-matrices in Appendix A.4.

## 4 Quantum Hamiltonian flow matching (QHFLOW)

### 4.1 Flow matching for Hamiltonian matrix

**Problem formulation.** Our goal is to train a model that predicts the Hamiltonian matrix $\mathbf{H}$ from a molecular configuration $\mathcal{M} = (\mathbf{x}, \mathbf{h})$. Here, for a molecule $\mathcal{M}$ with $M$ atoms, $\mathbf{x} \in \mathbb{R}^{M\times3}$ and

$\mathbf{h} \in \mathbb{R}^{M \times D}$ denotes the atom-wise coordinates and $D$-dimensional atom-wise features, respectively. We train the model using a atomic dataset $\mathcal{A}$ with a sample $(\mathcal{M}, \mathbf{H})$ corresponding to a molecular configuration $\mathcal{M}$ and its Hamiltonian matrix $\mathbf{H}$ obtained from conventional RH-DFT solvers.

**Conditional flow matching (CFM).** We first propose the flow matching for Hamiltonian matrix, by parameterizing our model as a continuous normalizing flow (CNF) model. Our CNF aims to learn a time-dependent vector field $v_t(\cdot|\mathcal{M})$ that outputs the Hamiltonian matrix $\mathbf{H}$ from solving the ODE $\frac{d}{dt}\mathbf{H}_t = v_t(\mathbf{H}_t|\mathcal{M})$ for time $t \in (0, 1]$. The ODE starts from $\mathbf{H}_0$ sampled from the prior distribution $p_0$ defined over the space $\mathbb{R}^{B \times B}$ of Hamiltonian matrices and outputs a Dirac distribution centered at the Hamiltonian matrix $\mathbf{H}$. To this end, we learn the vector field $v_t^\theta \approx v_t$ to match the distribution induced from the empirical distribution $(\mathcal{M}, \mathbf{H}_1) \sim \mathcal{A}$ associated with conditional probability paths $p_t(\mathbf{H}_t|\mathbf{H})$. Importantly, the conditional path $p_t(\mathbf{H}_t|\mathbf{H})$ is expressed using a closed-form conditional vector field $u_t(\cdot|\mathbf{H}_1)$: $\frac{d}{dt}\mathbf{H}_t = u_t(\mathbf{H}_t|\mathbf{H}_1, \mathcal{M})$ for $\mathbf{H}_0 \sim p_0$.

Following previous works [45, 35], we parameterize the conditional probability path using linear interpolation between $\mathbf{H}_0$ and $\mathbf{H}_1$, resulting in the conditional vector field $u_t(\mathbf{H}_t|\mathbf{H}_1, \mathcal{M}) = \frac{\mathbf{H}_1 - \mathbf{H}_t}{1-t}$. To train the CNF, we use the conditional flow matching objective defined as follows:

$$\mathcal{L}_{\text{CFM}} = \mathbb{E}_{(\mathbf{H},\mathcal{M})\sim\mathcal{A}, t\sim\mathcal{U}(0,1), \mathbf{H}_t\sim p_t(\cdot|\mathbf{H})} \left[ \left\| v_t^\theta(\mathbf{H}_t) - u_t(\mathbf{H}_t|\mathbf{H}_1, \mathcal{M}) \right\|_2^2 \right], \quad (5)$$

where $\mathcal{U}$ denotes an uniform distribution and $\|\cdot\|_2^2$ denotes the Frobeneous norm of a matrix. In practice, we parameterize the vector field by $v_t^\theta = \frac{\mathbf{H}_1^\theta(\mathbf{H}_t, \mathcal{M}) - \mathbf{H}_t}{1-t}$ using a neural network $\mathbf{H}_1^\theta(\cdot)$. This results in the final loss function:

$$\mathcal{L}_{\text{CFM}} = \mathbb{E}_{(\mathbf{H},\mathcal{M})\sim\mathcal{A}, t\sim\mathcal{U}(0,1), \mathbf{H}_t\sim p_t(\cdot|\mathbf{H})} \left[ \frac{1}{(1-t)^2} \left\| \mathbf{H}_1^\theta(\mathbf{H}_t, \mathcal{M}) - \mathbf{H}_{1,\mathcal{M}} \right\|_2^2 \right], \quad (6)$$

which one could interpret as training $\mathbf{H}_1^\theta$ to approximate the true Hamiltonian matrix $\mathbf{H}$ from its noisy versions $\mathbf{H}_t$. We provide a full description of the training and sampling algorithms in Appendix B.1.

**Fine-tuning for energy alignment.** In practice, many important quantum properties derived from DFT rely directly on orbital energies. To improve the accuracy of such downstream quantities including orbital energies, highest occupied molecular orbital (HOMO) energy, loweset unoccupied molecular orbital (LUMO) energy, and the HOMO–LUMO gap, we introduce a fine-tuning strategy that explicitly aligns the predicted orbital energies with their ground-truth DFT references. This approach is motivated by the weighted alignment loss (WALoss) introduced in WANet [25], which improves prediction quality by aligning the energy of predicted and reference matrix. Rather than incorporating this loss during full training, we adopt it as a post hoc fine-tuning objective.

Specifically, we define a fine-tuning objective that aligns the approximate orbital energies $\tilde{\epsilon}$ derived from our predicted Hamiltonian matrix $\mathbf{H}_1^\theta$ from CNF with the ground-truth orbital energies $\epsilon$ computed from RH-DFT as defined in Equation (2):

$$\mathcal{L}_{\text{FT}} = \mathbb{E}_{(\mathbf{H},\mathcal{M})\sim\mathcal{A}, t\sim\mathcal{U}(0,1), \mathbf{H}_t\sim p_t(\cdot|\mathbf{H})} \left[ \left\| \tilde{\epsilon}\left(\mathbf{H}_1^\theta(\mathbf{H}_t, \mathcal{M})\right) - \epsilon \right\|_2^2 \right] \quad (7)$$

Here, the ground-truth orbital energies $\epsilon$ are computed via the identity $\epsilon = \mathbf{C}^\top \mathbf{H} \mathbf{C}$, which follows from the orthonormality condition $\mathbf{C}^\top \mathbf{S} \mathbf{C} = \mathbf{I}$, where $\mathbf{C}$ is the ground-truth coefficient matrix and $\mathbf{S}$ is the overlap matrix. Following WALoss, we estimate $\tilde{\epsilon}$ using the predicted Hamiltonian: $\tilde{\epsilon} = \mathbf{C}^\top \mathbf{H}_1^\theta \mathbf{C} \approx (\mathbf{C}^\theta)^\top \mathbf{H}_1^\theta \mathbf{C}^\theta$, where $\mathbf{C}^\theta$ is the coefficient matrix obtained by solving the RH-DFT for the predicted Hamiltonian $\mathbf{H}_1^\theta$. We provide the implementation details in Appendix B.2.

### 4.2  SE(3)-equivariant flow matching with equivariant priors.

Here, we introduce our approach to design our QHFLOW to satisfy SE(3)-equivariance. We follow the framework of Song et al. [27] to learn an equivariant vector field under an invariant prior. To handle translational symmetry, our networks operate in a mean-free system by centering the molecular coordinates $\mathbf{X}$ at the origin, *i.e.*, set $\mathbf{X} \mapsto \mathbf{X} - \frac{1}{N}\mathbf{1}\mathbf{1}^\top\mathbf{X}$ given a vector $\mathbf{1} \in \mathbb{R}^{N \times 1}$ of ones. To make QHFLOW satisfy equivariance, we (1) parameterize the vector field $v_t^\theta$ using a SO(3)-equivariant neural network [23] and (2) propose *Gaussian orthogonal ensemble (GOE)* and *tensor expansion-based (TE)* priors for the Hamiltonian. The main challenge is to design new priors that are invariant to the rotation of the Hamiltonian matrix as described in Equation (4).

**Gaussian orthogonal ensemble (GOE) prior.** We here introduce a simple rotational invariant prior, the *GOE* prior. A GOE is defined over matrices $\mathbf{H} \in \mathbb{R}^{n \times n}$, with each element drawn from a Gaussian distribution: $H_{ij} \sim \mathcal{N}(0, \sigma^2)$. The log-density is proportional to the Frobenius norm of the matrix, which is invariant under any transformations associated with SO(3) irreps, *i.e.*, the 3D-rotation matrix ($\ell = 1$) and even the high-rank Wigner D-matrices ($\ell \geq 2$). Therefore, the prior satisfies the desired invariance, *i.e.*, $p(\mathbf{H}) = p(\mathcal{D}(\mathbf{R})\mathbf{H}\mathcal{D}(\mathbf{R})^{-1})$ for all $\mathbf{R} \in$ SO(3). We provide the full proof for the SO(3)-invariance in Appendix C.1.

**Tensor expansion-based (TE) prior.** We also introduce *TE* prior, an SO(3)-invariant distribution constructed by first sampling an irrep vector $\mathbf{w}^{(\ell)}$, then applying tensor expansion to produce a matrix $\mathbf{H}^{(\ell_1, \ell_2)} \in \mathbb{R}^{(2\ell_1+1) \times (2\ell_2+1)}$ that transforms covariantly under rotation $\mathbf{R} \in$ SO(3):

$$\mathbf{H}^{(\ell_1,\ell_2)} \xrightarrow{\mathbf{R}} D^{(\ell_1)}(\mathbf{R})\big(\mathbf{H}^{(\ell_1,\ell_2)}\big)D^{(\ell_2)}(\mathbf{R})^{-1}. \tag{8}$$

To this end, we first design an SO(3)-invariant distribution of irrep vector $p(\mathbf{w}^{(\ell_3)})$ by decomposing the vector into a radial and rotational parts: $\mathbf{w}^{(\ell)} = r D^{(\ell)}(\mathbf{R})\mathbf{w}^{(\ell_0)}$ where $r$ is a scalar norm and $\mathbf{w}^{(\ell_0)}$ is a fixed unit-norm irrep vector. *i.e.*, $Y^\ell(\hat{\mathbf{r}}_0)$ for fixed direction $\hat{\mathbf{r}}_0$. By sampling $\mathbf{r}$ independently of $\mathbf{R}$, and choosing $\mathbf{R}$ uniformly on SO(3), the resulting prior $p(\mathbf{w}^\ell)$ is SO(3)-invariant. In practice, we take $r \sim \text{Normal}(\mu, \sigma^2)$ to ensure positivity, and $\mathbf{R} \sim \text{Uniform}(\text{SO}(3))$.

Next, we apply tensor expansion, which maps a rank-$\ell$ irrep vector $\mathbf{w}^{(\ell)} \in \mathbb{R}^{(2\ell+1)}$ into a matrix of shape $(2\ell_1+1) \times (2\ell_2+1)$, indexed by the $(m_1, m_2)$ satisfying $-\ell_1 \leq m_1 \leq \ell_1$ and $-\ell_2 \leq m_2 \leq \ell_2$, respectively. Each entry of the resulting matrix is defined as follows:

$$\left(\bar{\otimes}\mathbf{w}^{(\ell)}\right)^{(\ell_1,\ell_2)}_{(m_1,m_2)} = \sum_{m=-\ell}^{\ell} C^{(\ell,m)}_{(\ell_1,m_1),(\ell_2,m_2)} w^{(\ell)}_m, \tag{9}$$

where $C$ denotes Clebsch-Gordan coefficients, $\bar{\otimes}$ denotes the tensor expansion operation. Note that the superscript $(\ell_1, \ell_2)$ indicates the output irrep structure and can be omitted when clear from context. The rotation of this expanded matrix is represented by the Wigner D-matrix as follows:

$$\left(\bar{\otimes}\mathbf{w}^{(\ell)}\right) \xrightarrow{\mathbf{R}} D^{(\ell_1)}(\mathbf{R})\left(\bar{\otimes}\mathbf{w}^{(\ell)}\right)D^{(\ell_2)}(\mathbf{R})^{-1}. \tag{10}$$

See Appendix C.2 for proof. This property allows us to construct SO(3)-invariant distributions over Hamiltonians. We formalize this in the following theorem:

**Theorem 1.** *Let* $\mathbf{H} = \big(\bar{\otimes}\mathbf{w}^{(\ell)}\big)^{(\ell_1,\ell_2)}$, *where an irrep vector* $\mathbf{w}^{(\ell)} \sim p(\mathbf{w}^{(\ell)})$ *is drawn from a SO(3)-invariant distribution. Then the induced distribution over* $\mathbf{H}$ *is invariant under SO(3) transformation:*

$$p(\mathbf{H}) = p(D^{(\ell_1)}(\mathbf{R})\mathbf{H}D^{(\ell_2)}(\mathbf{R})^{-1}), \qquad \forall \mathbf{R} \in SO(3). \tag{11}$$

We provide proof in Appendix C.3. Using the expansion, we can model the invariant distributions for each sub-matrix of Hamiltonian $\mathbf{H}$. We further extend this construction to build global invariant distributions over full Hamiltonian matrices $\mathbf{H}$ by composing multiple high-order irrep vector via tensor expansion. We provide a detailed construction of this multi-block formulation in Appendix C.4.

### 4.3 Model implementation

We construct QHFLOW by extending QHNet [23], an SE(3)-equivariant GNN for Hamiltonian prediction. Our architecture introduces time conditioning and two additional inputs: the current Hamiltonian $\mathbf{H}_t$ and the overlap matrix $\mathbf{S}$. The model takes $(\mathcal{M}, \mathbf{H}_t, \mathbf{S}, t)$ and predicts the target Hamiltonian $\mathbf{H}^\theta_1$ as described in Equation (6). We provided the implementation details in Appendix D.1.

**Feature intialization.** We first construct the atom-wise features $\mathbf{h}_i, \mathbf{s}_i$ based on (1) extracting atom-wise submatrices $\mathbf{H}_i, \mathbf{S}_i$ from $\mathbf{H}_t, \mathbf{S}$ and (2) mapping the submatrices into SO(3) irrep vectors based on Wigner-Eckart projections [46], i.e., set $\mathbf{h}_i = \texttt{ProjBlock}(\mathbf{H}_i), \mathbf{s}_i = \texttt{ProjBlock}(\mathbf{S}_i)$ where $\texttt{ProjBlock}$ flattens and applies change of basis transformations to the input matrix. We further encode time $t$ through tensor field network (TFN) [47] layer and sinusoidal encoding $e_t = f_{\text{time}}(t)$, i.e., set $\boldsymbol{m}_i = \texttt{TFN}(\mathbf{x}_i, \mathbf{h}_i, e_t)$.

**Message-passing layers.** To propagate information between neighbors, our model updates the irrep vectors $\mathbf{h}_i$ and $\mathbf{s}_i$ using SO(3)-equivariant attention-based layers adapted from Equiformer [48]:

$$\mathbf{h}_i \leftarrow \texttt{EquiBlock}\Big(\mathbf{h}_i, \{(\mathbf{h}_j, d_{ij})\}_{j \in \mathcal{N}_i}, e_t\Big), \quad \mathbf{s}_i \leftarrow \texttt{EquiBlock}\Big(\mathbf{s}_i, \{(\mathbf{s}_j, d_{ij})\}_{j \in \mathcal{N}_i}, e_t\Big),$$

where $\mathcal{N}_i$ denotes the neighbors of atom $i$ and $d_{ij}$ is distance between atom $i$ and $j$. We then apply a channel mixing module, Mix, which linearly combines the set of irrep vectors by flattening and reordering them according to their corresponding irrep indices while preserving SE(3)-equivariance. This module update the $\mathbf{m_i}$ in the residual manner:

$$\boldsymbol{m}_i \leftarrow \text{TFN}\Big(\boldsymbol{m}_i + \text{Mix}(\mathbf{h}_i, \mathbf{s}_i, \boldsymbol{m}_i), \big\{(\boldsymbol{m}_j, d_{ij})\big\}_{j \in \mathcal{N}_i}, e_t\Big). \tag{12}$$

This update process is repeated for a fixed number of iterations, allowing the model to integrate geometric and physical information across the molecular graph.

**Hamiltonian reconstruction.** The atom-wise irrep vectors $\boldsymbol{m}_i$ are passed into two TFN modules to predict Hamiltonian components. The first $\text{TFN}_s$ computes atom-wise irreps $\mathbf{w}_{ii}$ using messages from neighboring atoms, while the next $\text{TFN}_p$ predicts pairwise irreps $\mathbf{w}_{ij}$ between atom $i$ and $j$:

$$\mathbf{w}_{ii} = \text{TFN}_s\big(\boldsymbol{m}_i, \{(\boldsymbol{m}_k, d_{ik})\}_{k \in \mathcal{N}_i}\big), \quad \mathbf{w}_{ij} = \text{TFN}_p\big(\boldsymbol{m}_i, \boldsymbol{m}_j, d_{ij}\big). \tag{13}$$

Each output consists of a set of irrep vectors, $\mathbf{w}_{ii} = \{\mathbf{w}_{ii}^{(\ell)}\}_{\ell \in L_i}$ and $\mathbf{w}_{ij} = \{\mathbf{w}_{ij}^{(\ell)}\}_{\ell \in L_i}$, where $L_i$ is the set of angular momentum $\ell$ corresponding to the atom $i$. Then Hamiltonian blocks are constructed via tensor expansion $\bar{\otimes}$ and learnable weights $F$:

$$\mathbf{H}_{ij}^{(\ell_1, \ell_2)} = \sum_{\ell \in L_i} F_{ij}^{(\ell_1, \ell_2, \ell)} \left(\bar{\otimes} \mathbf{w}_{ij}^{(\ell)}\right)^{(\ell_1, \ell_2)}, \tag{14}$$

where $\ell_1 \in L_i$ and $\ell_2 \in L_j$, and the case $i = j$ corresponds to diagonal block. These blocks are then assembled into the full Hamiltonian matrix $\hat{\mathbf{H}}$, which is structured into $K \times K$ sub-matrices corresponding to all pairwise combinations of the $K$ groups of basis functions. All operations in this reconstruction preserve SE(3)-equivariance, ensuring that both $\hat{\mathbf{H}}$ and the learned vector field $v_t^\theta$ remain equivariant under any SE(3) transformation of the molecular geometry.

# 5 Experiments

In this section, we present experimental results that demonstrate the effectiveness of QHFLOW. We evaluate the performance on a public benchmark and compare it with competitive baselines [19, 20, 23, 26] when metrics are available. We provide training settings and hyperparameters in Appendix E.

**Datasets.** We evaluate our model on two datasets: MD17 [49, 19] and QH9 [50, 51, 24]. MD17 contains DFT Hamiltonians along molecular dynamics trajectories for four small molecules, water (4,900 structures), ethanol (30,000), malondialdehyde (26,978), and uracil (30,000). The PBE exchange–correlation [52, 53] functional with a GTO basis set is used for DFT.

QH9 consists of two subsets: stable and dynamic-300k, each supporting in-distribution (id) and out-of-distribution (ood) splits. The stable includes 130,831 molecules with up to 29 atoms. The id split randomly samples molecules, while the ood split groups them by size. The dynamic-300k contains 2,998 molecules, each with 100 perturbed geometries. The geo split assigns geometries randomly across splits for each molecule; the mol split partitions at the molecule level to test on unseen molecular identities. All QH9 Hamiltonians are computed using B3LYP [54] with the def2-SVP basis set. We provide the details about the dataset in Appendix E.1.

**Evaluation metrics.** Our evaluation metrics include the mean absolute error (MAE) of the Hamiltonian $\mathbf{H}$, the MAE of occupied orbital energies ($\epsilon_{\text{occ}}$), and the similarity score of the orbital coefficient matrix ($\mathcal{S}_c$). To assess physical fidelity on QH9, we also report the energy of LUMO ($\epsilon_{\text{LUMO}}$), HOMO ($\epsilon_{\text{HOMO}}$), and HOMO–LUMO gap ($\epsilon_\Delta$). We provide details of the metrics in Appendix E.2.

## 5.1 Performance on MD17 dataset

We extensively evaluate QHFLOW on the widely used MD17 benchmark dataset to assess its accuracy in predicting Hamiltonian matrices in diverse molecular geometries. As shown in Table 1, we compare QHFlow with baseline models in four representative molecules, and it consistently outperformed all existing methods. With up to a 73% reduction in Hamiltonian prediction error, QHFlow demonstrates strong predictive performance. This improvement highlights its ability to capture fine-grained variations in quantum Hamiltonians resulting from subtle geometric changes, an essential capability for accurately modeling physical and chemical behavior.

Table 1: **MD17 benchmark.** Results shown in **bold** denote the best result in each column, whereas those that are underlined indicate the second best.

| Model | Water (3 atoms) | | | Ethanol (9 atoms) | | | Malondialdehyde (9 atoms) | | | Uracil (12 atoms) | | |
|---|---|---|---|---|---|---|---|---|---|---|---|---|
| | $H\downarrow$ $[\mu E_h]$ | $\epsilon_{occ}\downarrow$ $[\mu E_h]$ | $\mathcal{S}_c\uparrow$ [%] | $H\downarrow$ $[\mu E_h]$ | $\epsilon_{occ}\downarrow$ $[\mu E_h]$ | $\mathcal{S}_c\uparrow$ [%] | $H\downarrow$ $[\mu E_h]$ | $\epsilon_{occ}\downarrow$ $[\mu E_h]$ | $\mathcal{S}_c\uparrow$ [%] | $H\downarrow$ $[\mu E_h]$ | $\epsilon_{occ}\downarrow$ $[\mu E_h]$ | $\mathcal{S}_c\uparrow$ [%] |
| SchNOrb | 165.4 | 279.3 | **100.00** | 187.4 | 334.4 | **100.00** | 191.1 | 400.6 | 99.00 | 227.8 | 1760. | 90.00 |
| PhiSNet | 15.67 | 85.53 | **100.00** | 20.09 | 102.04 | 99.81 | 21.31 | 100.6 | 99.89 | 18.65 | 143.36 | 99.86 |
| QHNet* | 11.70 | 26.06 | **100.00** | 27.99 | 99.33 | 99.99 | 29.60 | 100.16 | 99.92 | 26.80 | 127.93 | 99.87 |
| SPHNet | 23.18 | 182.29 | **100.00** | 21.02 | 82.30 | **100.00** | 20.67 | 95.77 | **99.99** | 19.36 | 118.21 | **99.99** |
| **Ours** | **4.93** | **19.29** | **100.00** | **5.33** | **29.03** | **100.00** | **3.80** | **22.68** | **99.99** | **3.68** | **30.54** | **99.99** |

Table 2: **QH9 benchmark.** WA-FT implies finetuned model with WAloss. Results shown in **bold** denote the best result in each column, whereas those that are underlined indicate the second best.

| Dataset | Model | $H\downarrow[\mu E_h]$ | $\epsilon_{occ}\downarrow[\mu E_h]$ | $\mathcal{S}_c\uparrow[\%]$ | $\epsilon_{LUMO}\downarrow[\mu E_h]$ | $\epsilon_{HOMO}\downarrow[\mu E_h]$ | $\epsilon_{\Delta}\downarrow[\mu E_h]$ |
|---|---|---|---|---|---|---|---|
| QH9-stable (id) | QHNet* | 77.72 | 963.45 | 94.80 | 18257.34 | 1546.27 | 17822.62 |
| | WANet | 80.00 | 833.62 | 96.86 | - | - | - |
| | SPHNet | 45.48 | 334.28 | 97.75 | - | - | - |
| | **Ours** | **22.95** | 119.67 | 99.51 | 437.96 | 179.48 | 553.87 |
| | **Ours (WA-FT)** | 23.85 | **101.92** | **99.56** | **187.48** | **92.22** | **206.15** |
| QH9-stable (ood) | QHNet* | 69.69 | 884.97 | 93.01 | 25848.83 | 1045.99 | 25370.10 |
| | SPHNet | 43.33 | 186.40 | 98.16 | - | - | - |
| | **Ours** | **20.01** | 84.54 | 99.04 | 321.20 | 130.74 | 395.83 |
| | **Ours (WA-FT)** | 20.55 | **72.64** | **99.16** | **171.24** | **77.96** | **179.57** |
| QH9-dynamic (300k-geo) | QHNet* | 88.36 | 1170.50 | 93.65 | 23269.41 | 2040.06 | 22407.96 |
| | WANet | 74.74 | 416.57 | 99.68 | - | - | - |
| | SPHNet | 52.18 | 100.88 | 99.12 | - | - | - |
| | **Ours** | **25.94** | 103.11 | 99.59 | 425.18 | 175.18 | 547.33 |
| | **Ours (WA-FT)** | 27.12 | **89.03** | **99.65** | **136.63** | **84.17** | **154.68** |
| QH9-dynamic (300k-mol) | QHNet* | 121.39 | 5554.36 | 86.02 | 53505.09 | 4352.76 | 50424.86 |
| | SPHNet | 108.19 | 1724.10 | 91.49 | - | - | - |
| | **Ours** | **45.91** | 442.56 | 98.65 | 1344.68 | 479.71 | 1605.03 |
| | **Ours (WA-FT)** | 46.60 | **424.75** | **98.74** | **912.10** | **403.51** | **1047.88** |

## 5.2 Performance on QH9 dataset

We evaluate QHFLOW on the QH9 dataset, a more challenging benchmark that includes various molecular sizes and compositions. This setting requires the model to generalize effectively across chemical and geometric variations. We trained on four different data splits. As shown in Table 2, QHFLOW consistently outperforms all prior methods in all metrics. In particular, it shows significant improvements in properties sensitive to eigenvalues such as $\epsilon_{LUMO}$, $\epsilon_{HOMO}$, and $\epsilon_{\Delta}$, showing the benefits of the flow-based model in capturing a physically meaningful structure in Hamiltonians. These results demonstrate QHFLOW's strong potential as a surrogate model for DFT.

We also apply our energy fine-tuning stage to a pretrained QHFLOW using the weighted alignment loss (WAloss) [25] for an additional 60,000 steps, which we denote as WA-FT. As shown in Table 2, this improves predictions of orbital energies and their downstream task, confirming that energy alignment fine-tuning serves as an effective inductive bias. We compare training from scratch with the WALoss and flow matching objectives in Appendix F.1.

## 5.3 DFT acceleration performance

Here, we show that our QHFLOW can accelerate DFT through initializing the SCF iterations [13–15] using the predicted Hamiltonian matrix, replacing the conventional SCF initialization methods.

**Preliminaries on SCF method.** Starting from an initial Hamiltonian $\mathbf{H}^{(0)}$, each SCF iteration solves the generalized eigenvalue problem $\mathbf{H}^{(k)}\mathbf{C}^{(k)} = \mathbf{S}\mathbf{C}^{(k)}\epsilon^{(k)}$ to obtain the coefficients $\mathbf{C}^{(k)}$. The coefficients are then used to construct the density $\rho^{(k)}$ and subsequently update the Hamiltonian to $\mathbf{H}^{(k+1)}$ through the mapping $\mathbf{C}^{(k)} \xrightarrow{\mathcal{Q}} \rho^{(k)} \xrightarrow{\mathcal{H}} \mathbf{H}^{(k+1)}$, where $\xrightarrow{\mathcal{Q}}$ and $\xrightarrow{\mathcal{H}}$ denote the density construction and Hamiltonian update operators, respectively. The iteration continues until convergence.

---

*We use our re-trained version of QHNet, based on the publicly available implementation.

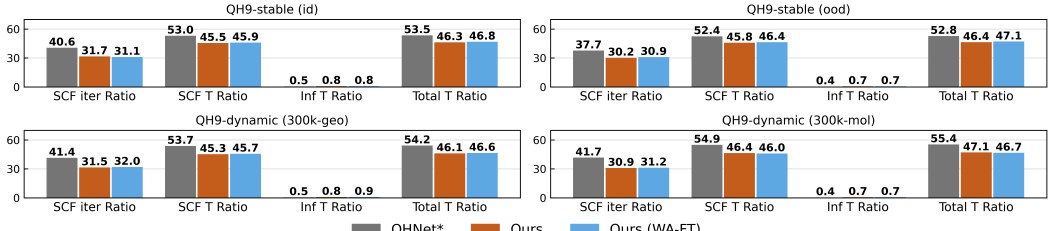

Figure 3: **DFT acceleration performance on 300 samples from the QH9 dataset.** All metrics are reported as percentages relative to conventional DFT (initialized with `minao`), which serves as the 100% baseline. The *SCF Iter Ratio* measures the ratio of SCF iterations required, while *Inf T Ratio*, *SCF T Ratio*, and *Total T Ratio* measure time. Lower *SCF Iter Ratio* and *Total T Ratio* values indicate faster convergence. For example, a *Total T Ratio* of 46% means QHFLOW converges in 46% of the conventional DFT time, including the negligible model inference time.

**DFT acceleration via SCF initialization.** We use QHFLOW's prediction $\hat{\mathbf{H}}$ to replace the conventional Hamiltonian guess by beginning the SCF loop closer to convergence. Performance is summarized with four relative metrics normalized by the conventional DFT baseline metric: SCF `Iter` Ratio: ratio of SCF iteration count, SCF `T` Ratio: ratio of SCF wall time, Inf T Ratio: ratio of inference time as a fraction of baseline SCF wall time, Total `T` Ratio: Ratio of net runtime after adding inference overhead. We provide the definition of metric in Appendix E.2.

We evaluate QHFLOW on 300 test molecules from the QH9 dataset, selected via a fixed index slice using a 3-step ODE-based sampling. As shown in Figure 3, QHFLOW significantly accelerates the convergence of SCF. For instance, on the QH9 `id` benchmark, QHFLOW achieves convergence in 46% of the runtime required by conventional DFT, corresponding to 54% relative speedup. Compared to previous ML-based initializations methods like QHNet which requires 53% of the conventional runtime, QHFLOW provides an additional reduction of 7% points. These results demonstrate the practical utility of QHFlow for accelerating real-world DFT simulations.

## 5.4 Ablation studies

Here, we ablate our design choices, with more results (*e.g.*, orbital energy metrics) in Appendix F.3.

**GOE, TE-based priors.** We explore the effects of the flow matching prior for Hamiltonian prediction by comparing the two priors introduced in Section 4: GOE and TE prior. The GOE prior treats each symmetric matrix as structure-free noise, whereas the TE prior injects group-theoretic bias that matches the blockwise symmetry of Hamiltonians.

Table 3 compares GOE and TE priors, where the TE prior results are identical to those of QHFlow reported in Table 2. We found that the TE prior consistently yields lower errors than the GOE prior across all splits. This highlights the importance of designing task-aligned and symmetry-aware priors for accurate Hamiltonian prediction.

Table 3: **Effect of prior distribution.**

| Data | Prior | $H\downarrow[\mu E_h]$ | $\epsilon_{\text{occ}}\downarrow[\mu E_h]$ | $\mathcal{S}_c\uparrow[\%]$ |
|------|-------|------|------|------|
| id | GOE | 25.93 | 154.65 | 99.39 |
|    | TE | **22.95** | **119.61** | **99.51** |
| ood | GOE | 20.41 | 87.32 | 98.95 |
|     | TE | **20.01** | **84.54** | **99.04** |
| geo | GOE | 29.39 | 122.14 | 99.49 |
|     | TE | **25.94** | **103.11** | **99.59** |
| mol | GOE | 46.78 | **419.68** | 98.65 |
|     | TE | **45.91** | 442.56 | **98.65** |

**Predictive variance.** To evaluate the robustness of QHFLOW to stochastic initialization, we repeat inference 5 times per molecule using the TE prior, changing only the seed that generates the initial noise sample. Results are shown in Table 4, with all metrics reported as `mean±std` over five runs. The predicted Hamiltonians exhibit a mean absolute deviation of just 0.03% and a standard deviation of predicted energy below $0.3\mu E_h$, much smaller than the prevalent chemical accuracy thresholds. These results confirm that QHFLOW produces stable and consistent predictions despite stochastic initialization.

Table 4: **Predictive variance of QHFLOW.**

| Data | Model | $H\downarrow[\mu E_h]$ | $\epsilon_{\text{occ}}\downarrow[\mu E_h]$ | $\mathcal{S}_c\uparrow[\%]$ |
|------|-------|------|------|------|
| id | Ours | $\mathbf{22.95}_{\pm 0.001}$ | $119.67_{\pm 0.211}$ | $99.51_{\pm 0.001}$ |
|    | WA-FT | $23.85_{\pm 0.001}$ | $\mathbf{101.92}_{\pm 0.279}$ | $\mathbf{99.56}_{\pm 0.002}$ |
| ood | Ours | $\mathbf{20.01}_{\pm 0.001}$ | $84.54_{\pm 0.007}$ | $99.04_{\pm 0.003}$ |
|     | WA-FT | $20.55_{\pm 0.002}$ | $\mathbf{72.64}_{\pm 0.018}$ | $\mathbf{99.16}_{\pm 0.006}$ |
| geo | Ours | $\mathbf{25.94}_{\pm 0.001}$ | $103.26_{\pm 0.031}$ | $99.59_{\pm 0.001}$ |
|     | WA-FT | $27.12_{\pm 0.002}$ | $\mathbf{89.03}_{\pm 0.213}$ | $\mathbf{99.65}_{\pm 0.001}$ |
| mol | Ours | $\mathbf{45.91}_{\pm 0.001}$ | $443.56_{\pm 0.171}$ | $98.65_{\pm 0.001}$ |
|     | WA-FT | $46.60_{\pm 0.001}$ | $\mathbf{424.75}_{\pm 0.324}$ | $\mathbf{98.74}_{\pm 0.001}$ |

# 6  Conclusion

In this work, we introduce QHFLOW, a high-order equivariant flow-based generative model designed to predict Hamiltonian matrices with high accuracy and physical consistency. By designing SE(3)-invariant priors GOE and TE, QHFLOW improves the accuracy and generalization across diverse geometries. Extensive experiments on the MD17 and QH9 datasets demonstrate that QHFLOW not only achieves state-of-the-art performance across multiple evaluation metrics but also significantly improves efficiency in downstream SCF calculations. Our results highlight the power of flow-based learning for scientific tasks and establish QHFLOW as a scalable and physically grounded framework for accelerating quantum chemistry simulations.

## Limitation

We did not fully verify the generalizability of our approach, *e.g.*, extension to new architectures (WANet [25], SPHNet [26]) or datasets (PubChemQH [25]), due to the lack of resources, public codes, and public datasets. Our flow-based formulation introduces computational overhead for solving ODEs. In the future, one could consider removing the overhead with the one-step generative models [55]. Our experiments are limited to gas-phase molecules with up to 29 atoms and two commonly used XC functionals (PBE and B3LYP). The algorithms are left to be verified for more general settings, such as periodic solids or higher-accuracy functionals (*e.g.*, hybrid or double-hybrid). We also acknowledge that our datasets consist of relatively small molecules. One could further test the scalability of our approach once new datasets on larger systems are available. We also note that our work does not give a theoretical explanation of how Hamiltonian prediction error translates to SCF acceleration. Developing a theoretically grounded algorithm specifically for SCF acceleration would be an interesting future research.

## Broader Impact

This work focuses on advancing machine learning methods for quantum chemistry applications and does not involve human subjects, personal data, or sensitive information. As such, we believe that there are no direct ethical concerns associated with this research. Nevertheless, we acknowledge that improved computational tools for molecular modeling could have downstream applications in sensitive areas such as pharmaceuticals or materials development. We encourage responsible and ethical use of our methods in accordance with applicable laws and scientific guidelines.

## Acknowledgements and Funding Disclosure

This work was partly supported by Institute for Information & communications Technology Planning & Evaluation(IITP) grant funded by the Korea government(MSIT) (RS-2019-II190075, Artificial Intelligence Graduate School Support Program(KAIST)), National Research Foundation of Korea(NRF) grant funded by the Ministry of Science and ICT(MSIT) (No. RS-2022-NR072184), GRDC(Global Research Development Center) Cooperative Hub Program through the National Research Foundation of Korea(NRF) grant funded by the Ministry of Science and ICT(MSIT) (No. RS-2024-00436165), and the Institute of Information & Communications Technology Planning & Evaluation(IITP) grant funded by the Korea government(MSIT) (RS-2025-02304967, AI Star Fellowship(KAIST)). We thank Minsu Kim, Hyomin Kim, and Hyosoon Jang for helpful discussions and feedback.

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

# A  Additional Preliminary

## A.1  Kohn-Sham density functional theory

Density functional theory (DFT) is a cornerstone of quantum chemistry and materials science, offering a computationally tractable framework for approximating the electronic structure of many-body systems. As directly solving the many-electron Schrödinger equation [37] is prohibitively expensive for systems with many particles, DFT reformulates the problem by expressing ground-state properties as a functional of the electron density $\rho(\mathbf{r})$. Since $\rho(\mathbf{r})$ depends only on three spatial coordinates, this significantly reduces the computational cost regardless of the number of electrons.

In practical applications, DFT is most commonly implemented via the Kohn–Sham formulation [1, 12], which introduces a fictitious system of non-interacting electrons designed to reproduce the true ground-state density of the real interacting system. The wavefunction of this non-interacting system, often referred to as the Kohn–Sham wavefunction $\Psi_{KS}$, is defined as follows. Since electrons are fermions, $\Psi_{KS}$ must obey the Pauli exclusion principle. This requires the wavefunction to be antisymmetric with respect to the exchange of any two electrons, a property that is achieved by constructing $\Psi_{KS}$ as a Slater determinant built from the $N$ single-particle Kohn-Sham orbitals $\{\psi_i\}_{i=1}^N$:

$$\Psi_{KS}(\mathbf{r}_1, \ldots, \mathbf{r}_N) = \frac{1}{\sqrt{N!}} \det[\psi_i(\mathbf{r}_j)]_{i,j=1}^N = \frac{1}{\sqrt{N!}} \begin{vmatrix} \psi_1(\mathbf{r_1}) & \psi_2(\mathbf{r_1}) & \cdots & \psi_N(\mathbf{r_1}) \\ \psi_1(\mathbf{r_2}) & \psi_2(\mathbf{r_2}) & \cdots & \psi_N(\mathbf{r_2}) \\ \vdots & \vdots & \ddots & \vdots \\ \psi_1(\mathbf{r_N}) & \psi_2(\mathbf{r_N}) & \cdots & \psi_N(\mathbf{r_N}) \end{vmatrix} \tag{15}$$

Originally, the electron density is calculated from the wavefunction $\Psi$:

$$\rho(\mathbf{r}) = N \int \cdots \int |\Psi(\mathbf{r}, \mathbf{r}_2, \ldots, \mathbf{r}_N)| d\mathbf{r}_2 \cdots d\mathbf{r}_N. \tag{16}$$

However, assuming orthonormal orbitals, the density expression simplifies to:

$$\rho(\mathbf{r}) = \sum_{i=1}^N |\psi_i(\mathbf{r})|^2. \tag{17}$$

Instead of explicitly constructing the full KS wavefunction, which scales exponentially with system size due to the determinant, we can directly obtain the electron density from the single-particle orbitals $\psi_i$. The single-particle orbitals $\{\psi_i(\mathbf{r})\}$ and their corresponding energies $\{\epsilon_i\}$ are found by solving the Kohn-Sham (KS) equations:

$$\mathcal{H}[\rho]\psi_i(\mathbf{r}) = \left[ -\frac{1}{2}\nabla^2 + V_{\text{ext}}(\mathbf{r}) + V_{\text{H}}[\rho](\mathbf{r}) + V_{\text{xc}}[\rho](\mathbf{r}) \right] \psi_i(\mathbf{r}) = \epsilon_i \psi_i(\mathbf{r}), \tag{18}$$

Here, the entire term in brackets is the effective KS Hamiltonian $\mathcal{H}[\rho]$, which is a functional of the electron density $\rho$. It consists of the Laplacian operator ($\nabla^2$) for the kinetic energy of the non-interacting electrons, the external potential ($V_{\text{ext}}$) from the atomic nuclei, the Hartree potential ($V_{\text{H}}$) representing classical electrostatic repulsion, and the crucial exchange-correlation potential ($V_{\text{xc}}$), which encapsulates all complex many-body quantum effects.

The ground-state electron density $\rho(\mathbf{r})$ is then constructed from the $N$ occupied KS orbitals obtained from solving these equations:

$$\rho(\mathbf{r}) = \sum_{i=1}^N |\psi_i(\mathbf{r})|^2. \tag{19}$$

It is important to note that the KS orbital $\psi_i$ is not, by itself, a physical wavefunction. It is a mathematical construct introduced solely to yield the same ground-state electron density as the real, interacting system. Additionally, the Hamiltonian $\mathcal{H}[\rho]$ depends on the density $\rho$, and $\rho$ itself is constructed from the orbitals $\psi_i$ that solve the KS equations, therefore forming a self-consistent problem.

## A.2 Group theory

We briefly introduce key concepts from group theory, providing the necessary background to understand the motivation behind equivariant designs in Hamiltonian prediction.

**Groups.** A *group* $G$ is a non-empty set equipped with a binary operation $\circ : G \times G \to G$, satisfying the following axioms:

- **Associativity**: For all $a, b, c \in G$, $(a \circ b) \circ c = a \circ (b \circ c)$.
- **Identity**: There exists an identity element $e \in G$ such that for all $a \in G$, $e \circ a = a \circ e = a$.
- **Inverse**: For each $a \in G$, there exists an inverse $a^{-1} \in G$ such that $a \circ a^{-1} = a^{-1} \circ a = e$.
- **Closure**: For all $a, b \in G$, the result $a \circ b$ is also in $G$.

We typically simplify notation by omitting $\circ$, writing $ab$ instead of $a \circ b$. Groups can be finite or infinite, discrete or continuous, compact or non-compact. Group theory [43] provides a rigorous mathematical foundation for describing symmetries in physical systems [56], an essential perspective for designing physically informed machine learning models.

**Group actions.** To make group theory applicable to structures in mathematics and physics, we define how groups act on sets. A group $G$ is said to act on a set $\mathcal{X}$ if there exists a function $\cdot_{\mathcal{X}} : G \times \mathcal{X} \to \mathcal{X}$[1] such that for all $g, h \in G$ and $x \in X$, the following conditions hold: (1) $e \cdot_{\mathcal{X}} x = x$, where $e$ is the identity element of $G$, and (2) $(gh) \cdot_{\mathcal{X}} x = g \cdot_{\mathcal{X}} (h \cdot_{\mathcal{X}} x)$. These conditions ensure that the group operation is compatible with its action on the set. In practice, such group actions often correspond to geometric transformations, such as translations, rotations, or reflections, on points or functions defined over Euclidean space.

**Equivariance.** When working with structured data such as molecular geometries or fields, it is often desirable that the operations we apply preserve the symmetries of the underlying space. A function $f : \mathcal{X} \to \mathcal{Y}$ is said to be *equivariant* with respect to group actions $\cdot_{\mathcal{X}}$ on the input space $\mathcal{X}$ and $\cdot_{\mathcal{Y}}$ on the output space $\mathcal{Y}$, if it satisfies

$$f(g \cdot_{\mathcal{X}} x) = g \cdot_{\mathcal{Y}} f(x), \quad \forall g \in G, x \in \mathcal{X}. \tag{20}$$

This means that applying a group transformation before or after the function yields consistent results. In physical systems, such as those governed by rotational symmetry (*e.g.*, SO(3) in molecules), equivariance ensures that rotated inputs yield correspondingly rotated outputs. This property is critical in designing models that generalize well across symmetrically equivalent configurations.

**Group representations.** Group actions on vector spaces are formalized through the notion of representations. A *representation* of a group $G$ on a vector space $V$ is a homomorphism $\rho : G \to \mathrm{GL}(V)$, where $\mathrm{GL}(V)$ denotes the group of invertible linear transformations on $V$. This mapping associates each group element with a linear operator that describes how the group acts on the space. The vector space $V$ is referred to as the *representation space*, and its dimension defines the representation's size.

In finite-dimensional real spaces $V = \mathbb{R}^d$, these representations are often expressed as invertible matrices $\mathbf{D}(g) \in \mathbb{R}^{d \times d}$. For instance, the group SO(3) of 3D rotations is represented by orthogonal matrices $g \in \mathrm{SO}(3)$ and its 3D rotation matrix $\mathbf{R} \in \mathbb{R}^{3 \times 3}$ acting on vectors $\mathbf{x} \in \mathbb{R}^3$ via matrix multiplication $\mathbf{D}(\mathbf{g})\mathbf{x} = \mathbf{R}\mathbf{x}$.

Two representations $\mathbf{D}(g)$ and $\mathbf{D}'(g)$ are said to be *equivalent* if they are related by a change of basis:

$$\mathbf{D}'(g) = \mathbf{Q}^{-1}\mathbf{D}(g)\mathbf{Q}, \tag{21}$$

for some invertible matrix $\mathbf{Q}$. Such equivalence indicates that the two representations describe the same group action under different coordinate systems.

## A.3 Irreducible representations

**Irreducible representations (Irreps).** A representation $\rho : G \to \mathrm{GL}(V)$ of a group $G$ on a vector space $V$ is called *irreducible* if it does not contain any nontrivial invariant subspace. More formally,

---

[1]The subscript of group action operator for indicating the input and output space is often omitted when clear from context, *i.e.,* $g \cdot_{\mathcal{X}} h = g \cdot h$.

a subspace $W \subseteq V$ is said to be invariant under the representation $\rho$ if for all group elements $g \in G$ and all vectors $w \in W$, the action satisfies:

$$\rho(g)w \in W. \tag{22}$$

Then, the representation $\rho$ is irreducible if the only invariant subspaces it admits are the trivial subspace $\{0\}$ and the entire representation space $V$.

Irreducible representations play a fundamental role because any finite-dimensional unitary representation of a compact group can be uniquely decomposed into a direct sum of irreducible representations, up to isomorphism [57]. This decomposition is foundational in analyzing the symmetry properties and constructing equivariant neural network architectures.

For the rotation group SO(3), the irreps are completely characterized by non-negative integers $\ell \in \mathbb{N}_0$. Each irreducible representation $D^{(\ell)} : \mathrm{SO}(3) \to \mathbb{C}^{(2\ell+1) \times (2\ell+1)}$ has dimension $2l+1$ and is explicitly realized by the action on spherical harmonics $Y_m^l(\theta, \phi)$. For a function $f : S^2 \to \mathbb{C}$ can be expanded in the basis of spherical harmonics as:

$$f(\theta, \phi) = \sum_{l=0}^{\infty} \sum_{m=-l}^{l} f_{lm} Y_m^l(\theta, \phi), \tag{23}$$

where $f_{lm} \in \mathbb{C}$ is the complex coefficients of the spherical harmonics. The action of a rotation $\mathbf{R} \in \mathrm{SO}(3)$ on $f$ is defined as:

$$(\mathbf{R} \cdot f)(\theta, \phi) = f(\mathbf{R}^{-1} \cdot (\theta, \phi)) = \sum_{l=0}^{\infty} \sum_{m=-\ell}^{\ell} \sum_{m'=-\ell}^{\ell} D_{mm'}^{(\ell)}(\mathbf{R}) f_{\ell m'} Y_m^\ell(\theta, \phi), \tag{24}$$

where $D^{(\ell)}(\mathbf{R})$ is the Wigner D-matrix corresponding to the irreducible representation labeled by $\ell$, acting linearly on the vector of expansion coefficients $\mathbf{f}^{(\ell)} = (f_{-\ell}, f_{-\ell+1}, \ldots, f_\ell) \in \mathbb{C}^{(2\ell+1)}$. Explicitly, under rotation:

$$\mathbf{f}^{(\ell)} \xrightarrow{\mathbf{R}} D^{(\ell)}(\mathbf{R}) \mathbf{f}^{(\ell)}. \tag{25}$$

Thus, irreps serve as the fundamental building blocks in constructing functions and features that transform equivariantly under the action of SO(3).

**Clebsch–Gordan tensor product.** Given two irreps $D^{(\ell_1)}$, $D^{(\ell_2)}$ of SO(3), their tensor product $D^{(\ell_1)} \otimes D^{(\ell_2)}$ acts on the tensor product space and is generally reducible. It decomposes into a direct sum of irreps:

$$D^{(\ell_1)} \otimes_{\mathrm{CG}} D^{(\ell_2)} \cong \bigoplus_{\ell=|\ell_1-\ell_2|}^{\ell_1+\ell_2} D^{(\ell)}, \tag{26}$$

where the decomposition is governed by Clebsch–Gordan (CG) coefficients. CG coefficients enable a basis transformation from the product space to irreps, playing crucial roles in quantum angular momentum coupling and equivariant neural networks.

Explicitly, for vectors $\mathbf{u}^{(\ell_1)} \in \mathbb{C}^{(2l_1+1)}$ and $\mathbf{v}^{(\ell_2)} \in \mathbb{C}^{(2\ell_2+1)}$, their CG tensor product produces an irreducible feature $\mathbf{w}^{(\ell)} \in \mathbb{C}^{(2\ell+1)}$:

$$w_m^{(\ell)} = \sum_{m_1, m_2} C_{(\ell_1, m_1), (\ell_2, m_2)}^{(\ell, m)} u_{m_1}^{(\ell_1)} v_{m_2}^{(\ell_2)}. \tag{27}$$

This construction ensures that $\mathbf{w}^{(\ell)}$ transforms correctly under $\mathbf{R} \in \mathrm{SO}(3)$:

$$\left( D^{(\ell_1)}(\mathbf{R}) \mathbf{u}^{(\ell_1)} \right) \otimes_{\mathrm{CG}} \left( D^{(\ell_2)}(\mathbf{R}) \mathbf{v}^{(\ell_2)} \right) = D^{(\ell)}(\mathbf{R}) \mathbf{w}^{(\ell)}. \tag{28}$$

Thus, CG tensor products ensure equivariant transformations, fundamental to maintaining consistency and symmetry in equivariant neural architectures.

**Tensor field networks (TFN).** The TFN [47] is a widely used architecture that achieves SE(3)-equivariance by explicitly encoding features as irreps of SO(3) and using spherical harmonics to process directional information. Each feature vector at a node is decomposed into spherical tensor components $V^{(\ell)} \in \mathbb{C}^{(2\ell+1)}$, where $\ell$ denotes the order of the irreducible representation. The key

component of the TFN layer is equivariant convolution and self-interaction. For the equivariant convolution, the filter function is generated by the spherical harmonic functions $Y_m^\ell(\hat{\mathbf{r}}_{ij})$ are applied to the direction between atoms $i$ and $j$, i.e., $\hat{\mathbf{r}}_{ij} = (\mathbf{r}_i - \mathbf{r}_j)/\|\mathbf{r}_i - \mathbf{r}_j\|$, and are modulated by a learnable radial function $R(\|\mathbf{r}_{ij}\|)$, implemented as an MLP to produce the filter:

$$F^{(\ell_{\text{in}}, \ell_f)}(\mathbf{r}_{ij}) = R^{(\ell_{\text{in}}, \ell_f)}(\|\mathbf{r}_{ij}\|)Y_m^{\ell_f}(\hat{\mathbf{r}}_{ij}). \tag{29}$$

The equivariant convolution aggregates information from neighboring nodes set $\mathcal{N}$ using the tensor product between the filters and neighboring features:

$$\tilde{V}_i^{(\ell_{\text{out}})} = \sum_{j \in \mathcal{N}} \left( F_c^{(\ell_{\text{in}}, \ell_f)}(\mathbf{r}_{ij}) \otimes V_j^{(\ell_{\text{in}})} \right)^{(\ell_{\text{out}})}, \tag{30}$$

where $\ell_{\text{out}} \in \{|\ell_{\text{in}} - \ell_f|, \ldots, (\ell_{\text{in}} + \ell_f)\}$. This convolution ensures that the result transforms equivariantly under rotation.

The self-interaction step applies a linear transformation over the channel dimension to each irreps independently:

$$\tilde{V}_{icm}^{(\ell)} = \sum_{c'} w_{cc'} V_{ic'm}^{(\ell)}, \tag{31}$$

preserving equivariance due to the linearity and diagonal structure in the irrep indices. By construction, all operations in TFNs are equivariant to SE(3), allowing them to model tensorial and directional quantities while respecting physical symmetries. This makes them especially well-suited for applications in molecular modeling, quantum chemistry, and materials science, where equivariance is crucial for generalization and physical accuracy.

### A.4 Symmetry and equivariance of RH-DFT

**Spherical harmonics and atomic orbital basis.** To numerically solve the Kohn–Sham equations, electronic wavefunctions are typically expanded in a basis of atomic orbitals. These orbitals are constructed using spherical harmonics $Y_m^\ell(\theta, \phi)$, which form a complete orthonormal basis on the sphere $\mathbb{S}^2$ and transform under the irreducible representations $D^{(\ell)}$ of the rotation group SO(3). An atomic orbital basis function takes the form [58]:

$$\phi_{n\ell m}(\mathbf{r}) = R_{n\ell}(\|\mathbf{r}\|)Y_m^\ell(\theta, \phi), \tag{32}$$

where $R_{n\ell}(\|\mathbf{r}\|)$ is a radial function (e.g., Gaussian [41] or Slater-type [40, 15]), and $Y_m^\ell$ captures the angular component. Here, $n$ is the principal quantum number, $\ell$ is the orbital angular momentum quantum number, and $m \in \{-\ell, \ldots, \ell\}$ is the magnetic quantum number [44].

These basis functions are central to Kohn–Sham DFT calculations, as molecular orbitals are expressed as linear combinations of atomic orbitals:

$$\psi_i(\mathbf{r}) = \sum_{\alpha = (n\ell m)} C_{\alpha i}\, \phi_{n\ell m}(\mathbf{r} - \mathbf{r}_\alpha), \tag{33}$$

where $\mathbf{C} \in \mathbb{R}^{B \times O}$ is the orbital coefficient matrix, $B$ is the number of basis functions, and $O$ is the number of occupied orbitals.

**Block Hamiltonian matrix.** The Hamiltonian matrix in RH-DFT can be decomposed according to the angular momentum quantum numbers $\ell$ grouping associated with the atomic orbitals. Specifically, let $\mathbf{H}^{(\ell, \ell')} \in \mathbb{C}^{(2\ell+1) \times (2\ell'+1)}$ denote the submatrix representing the coupling between two angular momentum numbers $\ell$ and $\ell'$. Under a rotation $\mathbf{R} \in$ SO(3), each such sub-matrix $\mathbf{H}^{(\ell, \ell')}$ transforms equivariantly according to the SO(3) irreps as follows:

$$\mathbf{H}^{(\ell, \ell')}[\mathcal{D}(\mathbf{R})\mathbf{C}] = D^{(\ell)}(\mathbf{R})\big(\mathbf{H}^{(\ell, \ell')}[\mathbf{C}]\big)D^{(\ell')}(\mathbf{R})^{-1}, \tag{34}$$

where atomic orbitals are grouped by their $\ell$ and $\ell'$, $\mathcal{D}(\mathbf{R})$ is block diagonal Wigner-D matrix, and $D^\ell$ and $D^{\ell'}$ are Wigner D-matrices for $\ell$ and $\ell'$, respectively. These submatrices $\mathbf{H}^{(\ell, \ell')}$, called orbital-wise Hamiltonian blocks, are the minimal building blocks of the full Hamiltonian that respect SO(3)-equivariance and have $(2\ell + 1) \times (2\ell' + 1)$ elements which corresponds to the combination of the magnetic quantum number $m$ and $m'$.

**Full Hamiltonian matrix.** In practice, basis functions are grouped not only by their angular momentum, but also by the atom they are centered on. For atom $i$, we denote its angular momentum support as $L_i = \{\ell_1, \ell_2, \ldots\}$, where each $\ell_k$ corresponds to its atomic shell (*e.g.*, $s$, $p$, $d$) represented in the basis set.

From this atomic partitioning, the Hamiltonian can be reorganized into block form where each submatrix $\mathbf{H}_{ij}^{(\ell,\ell')}$ corresponds to the interaction between angular momentum $\ell \in L_i$ of atom $i$ and $\ell' \in L_j$ of atom $j$. The full Hamiltonian $\mathbf{H}$ is thus a structured matrix composed of $K \times K$ blocks, where $K = \sum_i |L_i|$ is the total number of irreps (angular momentum groups) across all atoms, where $|L_i|$ denotes the cardinality of the set.

To construct the full Hamiltonian matrix, we first form the atom-pair matrix $\mathbf{H}_{ij}$ for every atom-pair $(i,j)$. This atom-pair matrix gathers all interactions between the angular momentum orbital groupings on the two atoms:

$$\mathbf{H}_{ij} = \begin{bmatrix} \mathbf{H}_{ij}^{(\ell_0,\ell_0')} & \mathbf{H}_{ij}^{(\ell_0,\ell_1')} & \cdots \\ \mathbf{H}_{ij}^{(\ell_1,\ell_0')} & \mathbf{H}_{ij}^{(\ell_1,\ell_1')} & \cdots \\ \vdots & \vdots & \ddots \end{bmatrix}, \qquad \ell_k \in L_i, \quad \ell_k' \in L_j, \tag{35}$$

where $\mathbf{H}_{ij}^{(\ell,\ell')} \in \mathbb{C}^{(2\ell+1) \times (2\ell'+1)}$ denotes the interaction between angular momentum $\ell \in L_i$ of atom $i$ and $\ell' \in L_j$ of atom $j$.

Next, we place these atom-pair blocks into a global block matrix to obtain the full RH Hamiltonian matrix: The full Hamiltonian matrix $\mathbf{H} \in \mathbb{C}^{n \times n}$, where $n = \sum_\ell (2\ell + 1)$, is assembled by concatenating the atom-pair blocks:

$$\mathbf{H} = \begin{bmatrix} \mathbf{H}_{ii} & \mathbf{H}_{ji} & \cdots \\ \mathbf{H}_{ij} & \mathbf{H}_{jj} & \cdots \\ \vdots & \vdots & \ddots \end{bmatrix}, \quad n = \sum_{i \in \mathcal{M}} \sum_{\ell \in L_i} (2\ell + 1), \tag{36}$$

for every atom-pair $(i,j)$ in the molecule $\mathcal{M}$.

For better intuition, we illustrate a schematic structure of the Hamiltonian matrix in Figure 4a. Here, blocks corresponding to $s$, $p$, and $d$ orbitals are arranged according to their angular momentum, highlighting the blockwise symmetry pattern induced by the underlying spherical harmonics basis.

**Rotational equivariance of full Hamiltonian.** The global action of a rotation $\mathbf{R}$ on the basis is encoded by the block-diagonal matrix as follows Equation (3):

$$\mathcal{D}(\mathbf{R}) = D^{(\ell_1)}(\mathbf{R}) \oplus \cdots \oplus D^{(\ell_K)}(\mathbf{R}) = \begin{bmatrix} D^{(\ell_1)}(\mathbf{R}) & 0 & \cdots \\ 0 & D^{(\ell_2)}(\mathbf{R}) & \cdots \\ \vdots & \vdots & \ddots \end{bmatrix}, \tag{37}$$

where each block $D^\ell(\mathbf{R})$ corresponds to the Wigner D-matrix for angular momentum $l$. The Hamiltonian matrix as a whole then transforms under rotation as:

$$\mathbf{H}[\mathbf{C}] \xrightarrow{\mathbf{R}} \mathbf{H}[\mathbf{R} \cdot \mathbf{C}] = \mathbf{R} \cdot \mathbf{H}[\mathbf{C}] = \mathcal{D}(\mathbf{R}) \, \mathbf{H}[\mathbf{C}] \, \mathcal{D}(\mathbf{R})^{-1}. \tag{38}$$

This global equivariance condition guarantees that the Hamiltonian transforms consistently with molecular rotations, preserving the physical structure of the system. Leveraging this symmetry is critical for constructing SE(3)-equivariant machine learning models, which not only respect physical laws but also improve model generalization and interpretability in quantum chemistry tasks. For better intuition, we illustrate a schematic structure of the Wigner D-matrix in Figure 4b.

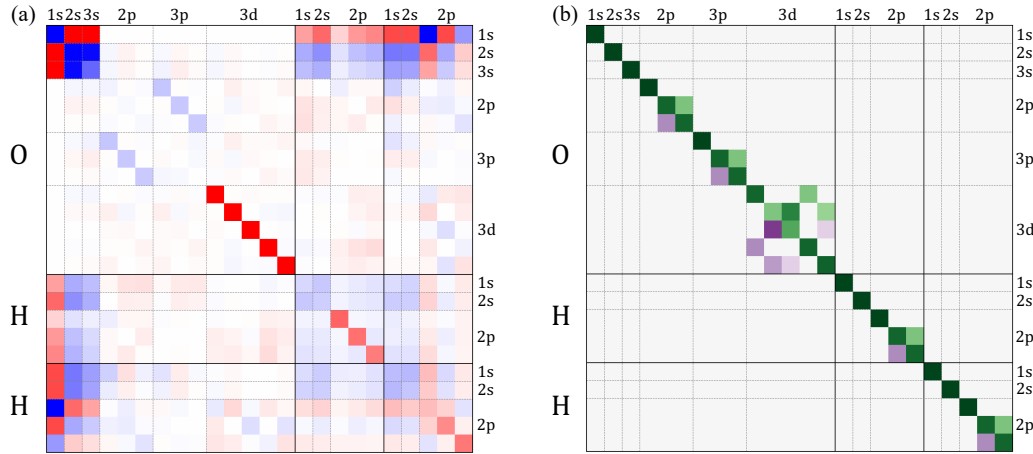

Figure 4: (a) Schematic illustration of the full Hamiltonian matrix $\mathbf{H}$ for a water molecule ($H_2O$). Color intensity indicates the magnitude of matrix elements, with red representing larger values and blue representing smaller values. (b) Schematic illustration of the full Wigner D-matrix $\mathcal{D}$ corresponding to $\mathbf{H}$, where green denotes larger values and purple denotes smaller values. Gray solid and dashed lines separate molecular blocks and orbital blocks, respectively, corresponding to submatrices defined by atomic orbital pairs.

# B  Training and sampling algorithms, and fine-tuning objective

## B.1  Training and sampling algorithms

We investigate two types of prior distributions for the initial Hamiltonian $\mathbf{H}_0$: Gaussian orthogonal ensemble (GOE) and tensor expansion-based (TE) priors. For the GOE prior, we sample each matrix entry independently as $M_{ij} \sim \mathcal{N}(0, \sigma^2)$, where we set $\sigma^2 = 1.0$ for MD17 and $\sigma^2 = 0.1$ for QH9.

For TE prior, we sample each irreducible component $\mathbf{w}^l$ by drawing its radial norm from $\mathrm{LogNormal}(1, \sigma^2 = 0.1)$ and applying a uniform SO(3) rotation to construct the full equivariant basis, which is then mapped to the Hamiltonian space via tensor expansion.

These choices provide flexibility in modeling different types of initial distributions, depending on the target dataset and symmetry constraints. Formal definitions and proofs of equivariance for both priors are provided in Appendix C.1 and Appendix C.4.

The training and sampling procedures based on the above rotationally invariant priors are summarized in Algorithm 1 and Algorithm 2, respectively.

---

**Algorithm 1** QHFlow training procedure

---

**Require:** Dataset of molecular configurations $\{\mathcal{M}_i\}$, target Hamiltonians $\mathbf{H}_{1,\mathcal{M}}$, overlap matrix $\mathbf{S}_{\mathcal{M}}$, model $f_\theta$
1: **for** each training step **do**
2:     Sample minibatch $\mathcal{B} = \{\mathcal{M}_i\}_{i=1}^B$
3:     **for** each $\mathcal{M}_i$ in $\mathcal{B}$ **do**
4:         Sample initial Hamiltonian $\mathbf{H}_0 \sim p_0$ and time $t \sim \mathcal{U}(0, 1)$
5:         Compute interpolated Hamiltonian: $\mathbf{H}_{t,\mathcal{M}} = (1 - t)\mathbf{H}_0 + t\mathbf{H}_{1,\mathcal{M}}$
6:         Predict $\mathbf{H}_{1,\mathcal{M}}^{(\theta)} = f_\theta(\mathbf{H}_{t,\mathcal{M}}, \mathbf{S}_{\mathcal{M}}, \mathcal{M}, t)$
7:         Compute flow matching loss: $\mathcal{L}_i = \|\mathbf{H}_{1,\mathcal{M}}^{(\theta)} - \mathbf{H}_{1,\mathcal{M}}\|^2$
8:     **end for**
9:     Compute $\mathcal{L} = \frac{1}{B}\sum_{i=1}^B \mathcal{L}_i$, update $\theta \leftarrow \theta - \eta\nabla_\theta\mathcal{L}$
10: **end for**

---

---

**Algorithm 2** QHFlow sampling procedure

---

**Require:** Molecular configuration $\mathcal{M}$, initial Hamiltonian $\mathbf{H}_0 \sim p_0$, model $f_\theta$, time discretization $\{t_0, t_1, \ldots, t_K\}$ where $t_0 = 0$ and $t_K = 1$
1: Initialize $\mathbf{H}_{t_0} = \mathbf{H}_0$
2: **for** $k = 0$ to $K - 1$ **do**
3:     Predict target Hamiltonian: $\mathbf{H}_{1,\mathcal{M}}^{(\theta)} \leftarrow f_\theta(\mathbf{H}_{t_k}, \mathbf{S}, \mathcal{M}, t_k)$
4:     Compute conditional vector field: $v_{t_k,\theta} \leftarrow (\mathbf{H}_{1,\mathcal{M}}^{(\theta)} - \mathbf{H}_{t_k})/(1 - t_k)$
5:     Update Hamiltonian: $\mathbf{H}_{t_{k+1}} \leftarrow \mathbf{H}_{t_k} + (t_{k+1} - t_k) \cdot v_{t_k,\theta}$
6: **end for**
7: Output $\hat{\mathbf{H}}_1 = \mathbf{H}_{t_K}$

---

These algorithms ensure that training aligns the predicted vector field with the true conditional flow, while inference produces a final Hamiltonian via integration through the learned ODE trajectory. This design ensures SE(3)-equivariance and physical consistency throughout the training and prediction processes.

## B.2  Fine-tuning objective

After pre-training with the flow matching objective, we further fine-tune QHFlow to enhance its spectral accuracy by introducing energy alignment objectives. To implement the energy alignment objective, we adopt the weighted alignment loss (WALoss) from WANet [25].

**Property of the RH-equation.** The relationship between the Hamiltonian matrix $\mathbf{H}$, the coefficient matrix $\mathbf{C}$, and the orbital energy matrix $\boldsymbol{\epsilon}$ is governed by the Roothaan–Hall equation:

$$\mathbf{HC} = \mathbf{SC}\boldsymbol{\epsilon}, \tag{39}$$

where $\mathbf{S}$ is overlap matrix. Using orthonormality condition $\mathbf{C}^\top \mathbf{SC} = \mathbf{I}$, this relation simplifies to:

$$\mathbf{C}^\top \mathbf{HC} = \boldsymbol{\epsilon}. \tag{40}$$

This identity forms the foundation for designing spectral alignment objectives.

**Weighted alignment loss (WALoss).** WALoss encourages the predicted Hamiltonian $\hat{\mathbf{H}}$ to match the spectral structure of the converged SCF Hamiltonian $\mathbf{H}^\star$. Let $\mathbf{C}^\star$ and $\boldsymbol{\epsilon}^\star$ denote the eigenvectors and eigenvalues of $\mathbf{H}^\star$, respectively. We define WALoss as:

$$\mathcal{L}_{\text{WALoss}} = \left\| \operatorname{diag}\left( (\mathbf{C}^\star)^\top \hat{\mathbf{H}} \mathbf{C}^\star \right) - \boldsymbol{\epsilon}^\star \right\|_2, \tag{41}$$

where $\operatorname{diag}(\cdot)$ extracts the diagonal elements and $\|\cdot\|_2$ denotes the L2 norm.

To emphasize physically important states, we place larger weights on eigenvalues up to the Lowest unoccupied molecular orbital (LUMO) *i.e.*, the $k+1$ lowest-energy orbitals. By projecting $\hat{\mathbf{H}}$ onto the fixed eigenbasis $\mathbf{C}^\star$, WALoss promotes alignment with the ground-truth spectrum. However, it is important to note that $(\mathbf{C}^\star)^\top \hat{\mathbf{H}} \mathbf{C}^\star$ does not exactly diagonalize $\hat{\mathbf{H}}$ unless $\hat{\mathbf{H}} = \mathbf{H}^\star$. Nevertheless, minimizing this discrepancy helps guide $\hat{\mathbf{H}}$ toward the correct eigenspectrum, improving orbital energy prediction and SCF behavior.

**Overall fine-tuning objective.** Our final finetuning objective combines the flow matching loss with WALoss:

$$\mathcal{L}_{\text{total}} = \mathcal{L}_{\text{CFM}} + \lambda_{\text{WA}} \mathcal{L}_{\text{WA}}, \tag{42}$$

where $\lambda_{\text{WA}}$ is the weighting coefficients controlling the contributions of the energy alignment terms, and we chose $\lambda_{\text{WA}} = 2.0$ for the finetuning.

# C Proofs and additional details

## C.1 Proof of SE(3)-equivariance of GOE

Let $\mathbf{M} \in \mathbb{R}^{n \times n}$ be a symmetric matrix drawn from the Gaussian orthogonal ensemble (GOE), characterized by:

$$\mathbb{E}[M_{ij}] = 0, \quad \mathrm{Var}(M_{ij}) = \sigma^2, \quad \text{with } M_{ij} = M_{ji}, \text{ and independent entries for } i \leq j. \quad (43)$$

Let $\mathbf{D} = \mathcal{D}(\mathbf{R})$ denote an orthogonal matrix representing a Wigner D-transformation corresponding to a rotation $\mathbf{R} \in \mathrm{SO}(3)$, satisfying orthogonality *i.e.* $\mathbf{D}\mathbf{D}^\top = \mathbf{I}$. We define the rotated matrix as:

$$\mathbf{M}' = \mathbf{D}\mathbf{M}\mathbf{D}^\top, \quad \text{with } M'_{ij} = \sum_{k,l} D_{ik} M_{kl} D_{jl}. \quad (44)$$

We aim to show that the distribution of $\mathbf{M}'$ is the same as that of $\mathbf{M}$, i.e., that GOE is invariant under conjugation by orthogonal transformations:

$$\rho(\mathbf{M}) = \rho(\mathbf{D}\mathbf{M}\mathbf{D}^\top). \quad (45)$$

To establish this, it suffices to verify that the first and second moments of the entries in $\mathbf{M}'$ match those of GOE.

**Mean.** Since $\mathbb{E}[M_{kl}] = 0$ for all $k, l$, we have:

$$\mathbb{E}[M'_{ij}] = \sum_{k,l} D_{ik} D_{jl} \mathbb{E}[M_{kl}] = 0. \quad (46)$$

**Covariance.** We now compute the covariance of the entries $M'_{ab}$ and $M'_{cd}$:

$$\mathbb{E}[M'_{ab} M'_{cd}] = \mathbb{E}\left[ \left( \sum_{i,j} D_{ai} D_{bj} M_{ij} \right) \left( \sum_{k,l} D_{ck} D_{dl} M_{kl} \right) \right] \quad (47)$$

$$= \sum_{i,j,k,l} D_{ai} D_{bj} D_{ck} D_{dl} \, \mathbb{E}[M_{ij} M_{kl}]. \quad (48)$$

Using the independence and zero mean of GOE entries, we know:

$$\mathbb{E}[M_{ij} M_{kl}] = \sigma^2 \delta_{ik} \delta_{jl}. \quad (49)$$

Thus,

$$\mathbb{E}[M'_{ab} M'_{cd}] = \sigma^2 \sum_{i,j} D_{ai} D_{bj} D_{ci} D_{dj} \quad (50)$$

$$= \sigma^2 \left( \sum_i D_{ai} D_{ci} \right) \left( \sum_j D_{bj} D_{dj} \right)$$

$$= \sigma^2 (\mathbf{D}\mathbf{D}^\top)_{ac} (\mathbf{D}\mathbf{D}^\top)_{bd}$$

$$= \sigma^2 \delta_{ac} \delta_{bd}. \quad (51)$$

This matches the covariance structure of the original GOE matrix. Since both $\mathbf{M}$ and $\mathbf{M}'$ are jointly Gaussian with zero mean and identical covariances, we conclude:

$$\mathbf{D}\mathbf{M}\mathbf{D}^\top \sim \mathbf{M}, \quad \text{i.e., GOE is invariant under orthogonal conjugation.} \quad (52)$$

This proves that the GOE prior is SO(3)-invariant, even for the high-rank Wigner D-matrices $\ell \geq 2$.

## C.2 Proof of the SE(3)-equivariance of tensor expansion operation

Since the $\mathbf{w}^{(\ell)}$ is the irrep vector, the $m$-th entry $w_m^\ell$ transforms under rotation $\mathbf{R} \in SO(3)$ as:

$$w_m^{(\ell)} \xrightarrow{\mathbf{R}} \left(\mathbf{R} \cdot \mathbf{w}^{(\ell)}\right)_m = \left(\sum_{m'} D_{mm'}^{(\ell)}(\mathbf{R}) \, w_{m'}^{(\ell)}\right), \tag{53}$$

where $D^{(\ell)}(\mathbf{R}) \in \mathbb{C}^{(2\ell_1+1)\times(2\ell_2+1)}$ is the Wigner D-matrix.

By substituting above Equation (53) in Equation (9), the entry of rotated tensor expansion is

$$\left(\bar{\otimes}\mathbf{w}^{(\ell)}\right)_{(m_1,m_2)}^{(\ell_1,\ell_2)} \xrightarrow{\mathbf{R}} \left(\bar{\otimes}\left[\mathbf{R} \cdot \mathbf{w}^{(\ell)}\right]\right) = \sum_m C_{(\ell_1,m_1),(\ell_2,m_2)}^{(\ell,m)} \left[\sum_{m'} D_{mm'}^{(\ell)}(\mathbf{R}) w_{m'}^{(\ell)}\right], \tag{54}$$

and we can exchange the order of summation:

$$\left(\bar{\otimes}\mathbf{w}^{(\ell)}\right)_{(m_1,m_2)}^{(\ell_1,\ell_2)} \xrightarrow{\mathbf{R}} \sum_{m'} \left(\sum_m C_{(\ell_1,m_1),(\ell_2,m_2)}^{(\ell,m)} D_{mm'}^{(\ell)}(\mathbf{R})\right) w_{m'}^{(\ell)}. \tag{55}$$

Now we can use a crucial identity from representation theory:

**Theorem 2.** *The Clebsch–Gordan coefficients provide a basis change between the product representation $D^{(\ell_1)}(\mathbf{R}) \otimes D^{(\ell_2)}(\mathbf{R})$ and the irreducible representation $D^{(\ell)}(\mathbf{R})$:*

$$\sum_m C_{(\ell_1,m_1),(\ell_2,m_2)}^{(\ell,m)} D_{mm'}^{(\ell)}(\mathbf{R}) = \sum_{m_1',m_2'} D_{m_1 m_1'}^{(\ell_1)}(\mathbf{R}) D_{m_2 m_2'}^{(\ell_2)}(\mathbf{R}) C_{(\ell_1,m_1'),(\ell_2,m_2')}^{(\ell,m')}. \tag{56}$$

We can substitute the identity, then we have:

$$\left(\bar{\otimes}\mathbf{w}^{(\ell)}\right)_{(m_1,m_2)}^{(\ell_1,l_2)} \xrightarrow{\mathbf{R}} \sum_{m_1',m_2'} D_{m_1 m_1'}^{(\ell_1)}(\mathbf{R}) D_{m_2 m_2'}^{(\ell_2)}(\mathbf{R}) \left(\sum_{m'} C_{(\ell_1,m_1'),(\ell_2,m_2')}^{(\ell,m')} w_{m'}^{(\ell)}\right), \tag{57}$$

and the last sum implies the

$$\sum_{m'} C_{(\ell_1,m_1'),(\ell_2,m_2')}^{(\ell,m')} w_{m'}^{(\ell)} = \left(\bar{\otimes}\mathbf{w}^{(\ell)}\right)_{(m_1',m_2')}^{(\ell_1,\ell_2)} \tag{58}$$

Therefore,

$$\left(\bar{\otimes}\mathbf{w}^{(\ell)}\right)_{(m_1,m_2)}^{(\ell_1,\ell_2)} \xrightarrow{\mathbf{R}} \sum_{m_1',m_2'} D_{m_1 m_1'}^{(\ell_1)}(\mathbf{R}) D_{m_2 m_2'}^{(\ell_2)}(\mathbf{R}) \left(\bar{\otimes}\mathbf{w}^{(\ell_3)}\right)_{(m_1',m_2')}^{(\ell_1,\ell_2)}, \tag{59}$$

in the matrix form,

$$\left(\bar{\otimes}\mathbf{w}^{(\ell)}\right) \xrightarrow{\mathbf{R}} D^{(\ell_1)}(\mathbf{R}) \left(\bar{\otimes}\mathbf{w}^{(\ell)}\right) D^{(\ell_2)}(\mathbf{R})^{-1}. \tag{60}$$

## C.3 Proof of the invariance of the tensor expansion based (TE) prior

To construct an SO(3)-invariant distribution over matrices via tensor expansion, we begin with a rotationally invariant distribution over irreducible features $\mathbf{w}^{(\ell)} \in \mathbb{C}^{(2\ell+1)}$. We define this distribution by factorizing it into two independent components:

- A magnitude $r = \|\mathbf{w}^{(\ell)}\|$ drawn from a radial distribution, e.g., $r \sim \text{Normal}(1, \sigma^2)$.
- A direction sampled uniformly on the sphere $\mathbb{S}^2$, induced by a Haar-uniform rotation $\mathbf{R} \in SO(3)$, applied to a canonical unit-norm feature vector $\mathbf{w}_0^{(\ell)}$, such that $\mathbf{w}^{(\ell)} = r D^{(\ell)}(\mathbf{R}) \mathbf{w}_0^{(\ell)}$.

This construction ensures that $\mathbf{w}^{(\ell)}$ is distributed isotropically, i.e., $p(\mathbf{w}^{(\ell)}) = p(D^{(\ell)}(\mathbf{R})\mathbf{w}^{(\ell)})$ for any $\mathbf{R} \in SO(3)$. Now, from the tensor expansion defined as:

$$\mathbf{H}^{(\ell_1,\ell_2)} = \left(\bar{\otimes}\mathbf{w}^{(\ell)}\right)^{(\ell_1,\ell_2)}, \tag{61}$$

we know from the previous subsection that this construction transforms under SO(3) as:

$$\mathbf{H}^{(\ell_1,\ell_2)} \xrightarrow{\mathbf{R}} D^{(\ell_1)}(\mathbf{R})\mathbf{H}^{(\ell_1,\ell_2)}D^{(\ell_2)}(\mathbf{R})^{-1}. \tag{62}$$

To prove invariance, consider the pushforward distribution:

$$p(\mathbf{H}^{(\ell_1,\ell_2)}) = p(\mathbf{w}^{(\ell)}) = p(D^{(\ell)}(\mathbf{R})\mathbf{w}^{(\ell)}) \quad \forall \mathbf{R} \in SO(3). \tag{63}$$

Because $\mathbf{H}^{(\ell_1,\ell_2)}$ is a deterministic, equivariant function of $\mathbf{w}^{(\ell)}$, and $p(\mathbf{w}^{(\ell)})$ is rotation-invariant, the induced distribution $p(\mathbf{H}^{(\ell_1,\ell_2)})$ must also be invariant under conjugation by $D^{(\ell_1)}(\mathbf{R})$ and $D^{(\ell_2)}(\mathbf{R})$. Thus:

$$p(\mathbf{H}^{(\ell_1,\ell_2)}) = p(D^{(\ell_1)}(\mathbf{R})\,\mathbf{H}^{(\ell_1,\ell_2)}\,D^{(\ell_2)}(\mathbf{R})^{-1}). \tag{64}$$

This proves that the prior distribution constructed via tensor expansion from rotationally invariant $\mathbf{w}^{(\ell)}$ yields an SO(3)-equivariant (conjugation-invariant) distribution over matrix-valued outputs.

## C.4 Proof of the invariance of the multiple tensor expansion prior

We now generalize the single-irrep vector expansion to construct a full Hamiltonian matrix $\mathbf{H} \in \mathbb{R}^{n \times n}$ composed of multiple irrep vectors components. Let the atomic orbital basis consist of a set of irrep vectors indexed by $L = \{\ell_1, \ell_2, \ldots, \ell_B\}$, where each angular momentum $\ell_i$ corresponds to a subspace of dimension $(2\ell_i + 1)$.

For each irrep vector rank $\ell_i$, we define a random irrep vector $\mathbf{w}^{(\ell_i)} \in \mathbb{C}^{(2\ell_i+1)}$, sampled independently from a rotationally invariant distribution:

$$\mathbf{w}^{(\ell_i)} = r_i D^{(\ell_i)}(\mathbf{R}_i)\mathbf{w}_0^{(\ell_i)}, \quad r_i \sim \mathrm{LogNormal}(1, \sigma^2), \quad \mathbf{R}_i \sim \mathrm{Uniform}(SO(3)). \tag{65}$$

We then define a factorized prior:

$$p(\mathbf{w}^{(\ell_1)}, \ldots, \mathbf{w}^{(\ell_B)}) = \prod_{i=1}^{B} p_i(\mathbf{w}^{(\ell_i)}), \tag{66}$$

with each $p_i(\mathbf{w}^{(\ell_i)})$ being SO(3)-invariant.

Using the tensor expansion, each pair $(\ell_i, \ell_j)$ generates a block of the Hamiltonian matrix via:

$$\mathbf{H}^{(\ell_i,\ell_j)} = \bar{\otimes}_{\ell_i,\ell_j}\mathbf{w}^{(\ell_k)}, \quad \text{for some } \ell_k \in \{|\ell_i - \ell_j|, \ldots, \ell_i + \ell_j\}. \tag{67}$$

The full Hamiltonian is then assembled as:

$$\mathbf{H} = \bigoplus_{i,j} \mathbf{H}^{(\ell_i,\ell_j)}. \tag{68}$$

From the single-component proof, we know that each block transforms as:

$$\mathbf{H}^{(\ell_i,\ell_j)} \xrightarrow{\mathbf{R}} D^{(\ell_i)}(\mathbf{R})\mathbf{H}^{(\ell_i,\ell_j)}D^{(\ell_j)}(\mathbf{R})^{-1}. \tag{69}$$

Therefore, under a global rotation $\mathbf{R} \in SO(3)$, the full matrix $\mathbf{H}$ transforms as:

$$\mathbf{H} \xrightarrow{\mathbf{R}} \mathcal{D}(\mathbf{R})\,\mathbf{H}\,\mathcal{D}(\mathbf{R})^{-1}, \tag{70}$$

where $\mathcal{D}(\mathbf{R})$ is the block-diagonal rotation operator acting on the full basis.

Since each $\mathbf{w}^{(\ell_i)}$ is sampled independently from an SO(3)-invariant distribution, the joint distribution of all irreducible components $p(\mathbf{H})$ remains invariant under this global conjugation:

$$p(\mathbf{H}) = p\left(\mathcal{D}(\mathbf{R})\,\mathbf{H}\,\mathcal{D}(\mathbf{R})^{-1}\right), \quad \forall \mathbf{R} \in SO(3). \tag{71}$$

This completes the proof that the full Hamiltonian matrix constructed via multiple tensor expansions, using independently sampled spherical features from invariant priors, defines an SO(3)-invariant distribution over matrices.

# D Implementation details

## D.1 Model implementation

We build our model upon the official QHNet [23] and DEQHNet [59] codebases, which implement SE(3)-equivariant GNN for Hamiltonian matrix prediction. QHNet is selected as our backbone for its balance of architectural simplicity and strong predictive performance. Although more recent models such as WANet [25] and SPHNet [26] have demonstrated improvements, their codebases are not publicly available and thus are not considered here. Notably, our method is model-agnostic and could be paired with more expressive backbones if desired.

We extend QHNet by incorporating additional physically meaningful inputs: the current Hamiltonian matrix $\mathbf{H}_t$, the overlap matrix $\mathbf{S}$, the molecular configuration $\mathcal{M}$, and the time $t$. Unlike the original QHNet, which processes only $\mathcal{M}$, our model is designed to handle $\mathbf{H}_t$, and $\mathbf{S}$ as well, following symmetry-preserving strategies inspired by DEQHNet and OrbNet-Equi [60].

**Project Block.** To map the equivariant matrix features $\mathbf{H}$ and $\mathbf{S}$ into the SO(3) irrep vectors, we used *diagonal reduction* in OrbNet [60] based on the Wigner-Eckart theorem [46]. For atom $i$, we first extract the atom-wise matrices, $\mathbf{H}_i := \mathbf{H}_{ii}$ (see Equation (35)). Each element of $\mathbf{H}_i$ carries two orbital index, $(\ell, m)$ and $(\ell', m')$. We project these matrices onto rank-$\ell_o$ irrep vectors $\mathbf{h}_i^{(\ell_o)} = [h_i^{(\ell_o, -\ell_o)}, \dots h_i^{(\ell_o, \ell_o)}] \in \mathbb{C}^{(2\ell_o + 1)}$ by

$$h_i^{(\ell_o, m_o)} = \sum_{(\ell, \ell')} \sum_{(m, m')} \mathbf{H}_i^{(\ell, m; \ell', m')} Q_{i, (\ell_o, m_o)}^{(\ell, m; \ell', m')}, \tag{72}$$

where $\mathbf{H}_i^{(\ell, m; \ell', m')}$ is an element of submatrix $\mathbf{H}_i^{(\ell, \ell')}$ that corresponds to the $m$ and $m'$ index, and a set of atom-wise projection weights $Q_{i, (\ell_o, m_o)}^{(\ell, m; \ell', m')}$ which plays the role of Clebsch-Gordan-like projection weights. Because the map in Equation (72) is an SO(3) tensor contraction, the resulting feature vectors remain equivariant. The same procedure is applied to overlap matrix $\mathbf{S}$, giving two parallel streams of SE(3)-equivariant atomic features that are subsequently fused in the network.

**Construction of projection weights.** To construct the coefficients atom-wise projection coefficients, we compute three-center integrals involving orbital basis function $\Phi_i^{(\ell, m)}$ and auxiliary spherical-type basis functions $\tilde{\Phi}_i^{(l, m)}$:

$$Q_{i, (\ell_o, m_o)}^{(\ell, m; \ell', m')} = \int \left( \Phi_i^{(\ell, m)}(\mathbf{r}) \right)^* \Phi_i^{(\ell', m')}(\mathbf{r}) \tilde{\Phi}_i^{(\ell_o, m_o)}(\mathbf{r}) \, d\mathbf{r}. \tag{73}$$

In practice, the angular parts of these integrals are related with the Clebsch-Gordan coefficients due to their relation to spherical harmonics:

$$Q_{i, (\ell_o, m_o)}^{(\ell, m; \ell', m')} \propto \int_{\mathbb{S}^2} Y_m^\ell(\hat{\mathbf{r}}) Y_m^{\ell'}(\hat{\mathbf{r}}) \left( Y_{m_o}^{\ell_o}(\hat{\mathbf{r}}) \right)^* d\mathbf{r}. \tag{74}$$

## D.2 Training objective

Our training objective adopts a residual learning strategy, following prior works such as WANet and SHNet. Rather than directly predicting the converged Hamiltonian, the model learns to approximate the residual between the initial and converged Hamiltonians using a time-dependent vector field within the conditional flow matching framework. We define the residual target as done in prior works:

$$\mathbf{H}_{1, \mathcal{M}} := \mathbf{H}_{\mathcal{M}}^\star - \mathbf{H}_{\mathcal{M}}^{(0)}, \tag{75}$$

where $\mathbf{H}_{\mathcal{M}}^{(0)}$ is the initial guess of Hamiltonian matrix[2], and $\mathbf{H}_{\mathcal{M}}^\star$ is the SCF solution. The final prediction of Hamiltonian is obtained by summing the predicted residual and the initial Hamiltonian guess.

The conditional flow matching loss is defined as:

$$\mathcal{L} = \mathbb{E}_{(\mathbf{H}, \mathcal{M}) \sim \mathcal{A}, t \sim \mathcal{U}(0,1), \mathbf{H}_t \sim p_t} \left[ \left\| v_{t, \theta}(\mathbf{H}_{t, \mathcal{M}}) - u_t(\mathbf{H}_{t, \mathcal{M}} \mid \mathbf{H}_{1, \mathcal{M}}) \right\|_2^2 \right] \tag{76}$$

$$= \mathbb{E}_{(\mathbf{H}, \mathcal{M}) \sim \mathcal{A}, t \sim \mathcal{U}(0,1), \mathbf{H}_t \sim p_t} \left[ \frac{1}{(1-t)^2} \left\| \mathbf{H}_{1, \mathcal{M}}^{(\theta)} - \mathbf{H}_{1, \mathcal{M}} \right\|_2^2 \right], \tag{77}$$

---

[2]We use the `minao` initialization from the `PySCF` package, and this guess does not contains SCF steps.

where the second line follows from the analytical form of the conditional velocity field $u_t$, showing that the model is penalized based on the squared residual error at each time step. This formulation encourages smooth convergence from the initial guess to the target solution. To simplify the objective, we omit the $1/(1-t)^2$ time-scaling for our implementation.

# E  Experimental study settings

## E.1  Dataset preparation

To demonstrate the effectiveness of flow-matching-based training, we conduct experiments on two molecular datasets: MD17 and QH9. The MD17 represents a relatively simple task compared to the QH9, focusing solely on small systems and their conformational space. The PubChemQH9 is not considered since their dataset and codebase are not publicly released.

**MD17.** The MD17 [49, 19] dataset consists of quantum chemical simulations for four small organic molecules: water ($H_2O$), ethanol ($C_2H_5OH$), malondialdehyde ($CH_2(CHO)_2$), and uracil ($C_4H_4N_2O_2$). It provides a comprehensive set of molecular properties, including geometries, total energies, forces, Kohn–Sham Hamiltonian matrices, and overlap matrices. All reference computations were implemented via the ORCA electronic structure package [61] using the PBE exchange–correlation functional [52, 53] and the def2-SVP Gaussian-type orbital (GTO) basis set. We follow the standard data split protocol used in prior work [19, 20, 23] to divide each molecule's conformational data into training, validation, and test sets. The detailed dataset statistics are summarized in Table 5 and MOs in the table imply molecular orbitals (*i.e..* s,p,d,f)

Table 5: The statistics of MD17 dataset [19].

| Dataset | # of structures | Train | Val | Test | # of atoms | # of orbitals | # of occupied MOs |
|---------|---------|---------|---------|---------|---------|---------|---------|
| Water | 4,900 | 500 | 500 | 3,900 | 3 | 24 | 5 |
| Ethanol | 30,000 | 25,000 | 500 | 4,500 | 9 | 72 | 10 |
| Malondialdehyde | 26,978 | 25,000 | 500 | 1,478 | 9 | 90 | 19 |
| Uracil | 30,000 | 25,000 | 500 | 4,500 | 12 | 132 | 26 |

**QH9.** The QH9 dataset [24] is a large-scale quantum chemistry benchmark designed to support the training and evaluation of machine learning models for Hamiltonian matrix prediction across diverse chemical structures. It is based on the QM9 [50, 51] molecular dataset and includes 130,831 Hamiltonian matrices from stable molecular geometries, as well as 2,698 molecular dynamics trajectories. The dataset covers small organic molecules composed of up to nine heavy atoms (C, N, O, and F). All Hamiltonians were computed using the PySCF [62] quantum chemistry package with the B3LYP [54] exchange–correlation functional and the def2-SVP Gaussian-type orbital (GTO) basis set. We provide the detailed dataset statistics in Table 6.

QH9 is organized into two main subsets, QH9-`stable` and QH9-`dynamic-300k`, and provides four standard evaluation splits: stable-id, stable-ood, dynamic-300k-`geo`, and dynamic-300k-`mol`. The `id` split randomly partitions the QH9-`stable` subset into training, validation, and test sets, while the `ood` split is constructed based on molecular size. Specifically, it groups molecules with 3–20 atoms in the training set, molecules with 21–22 atoms in the validation set, and molecules with 23–29 atoms in the test set. This `ood` setup allows for a rigorous evaluation of out-of-distribution (OOD) generalization in terms of molecular complexity.

The `geo` and `mol` splits are based on molecular dynamics (MD) trajectories, where each molecule is associated with 100 geometry snapshots. In the `geo` split, geometries are randomly assigned within each molecule: 80 for training, 10 for validation, and 10 for testing. Thus, while all molecules are present across the splits, the specific conformations are disjoint, enabling an assessment of geometric generalization. In contrast, the `mol` split divides the 2,698 molecules themselves into disjoint sets with an 80/10/10 train/validation/test ratio. All 100 geometries of a molecule are assigned to the same subset. This split presents a more challenging task than the `geo` split, as it requires generalization to entirely unseen molecular identities rather than just new conformations.

## E.2  Evaluation Metrics

To comprehensively evaluate the performance of Hamiltonian prediction models, we adopt several metrics that measure accuracy, physical fidelity, and downstream impact on quantum chemical properties. Below are detailed descriptions of each metric used in our experiments.

Table 6: The statistics of QH9 dataset [24].

| Dataset | # of structures | # of Molecules | Train | Val | Test |
|---|---|---|---|---|---|
| Stable-id | 130,831 | 130,831 | 104,664 | 13,083 | 13,084 |
| Stable-ood | 130,831 | 130,831 | 104,001 | 17,495 | 9,335 |
| Dynamic-300k-geo | 299,800 | 2,998 | 239,840 | 29,980 | 29,980 |
| Dynamic-300k-mol | 299,800 | 2,998 | 239,800 | 29,900 | 30,100 |

**Hamiltonian MAE.** This metric measures the mean absolute error (MAE) between the predicted Hamiltonian matrix $\hat{\mathbf{H}}$ and the ground-truth matrix $\mathbf{H}^\star$:

$$\mathrm{MAE}(\mathbf{H}) = \frac{1}{n^2} \sum_{i,j=1}^{n} \left\| \hat{\mathbf{H}}_{ij} - \mathbf{H}_{ij}^\star \right\|_2^2, \tag{78}$$

where $n$ is the length of the Hamiltonian matrix ($\mathbf{H} \in \mathbb{R}^{n \times n}$), and $\|\cdot\|_2^2$ is the Frobenius norm. This metric directly reflects the quality of the predicted Hamiltonian in element-wise terms.

**Occupied orbital energy MAE ($\epsilon_{\mathbf{occ}}$).** This metric measures the MAE between the predicted and true orbital energies for only the occupied orbitals. Let $\hat{\epsilon}$ and $\epsilon^\star$ be predicted and the ground-truth generalized eigenvalues of the Hamiltonian, respectively, and let $\mathcal{I}_{\mathrm{occ}}$ be the set of indices of the occupied orbitals:

$$\mathrm{MAE}(\epsilon_{\mathrm{occ}}) = \frac{1}{\mathrm{card}(\mathcal{I}_{\mathrm{occ}})} \sum_{i \in \mathcal{I}_{\mathrm{occ}}} \|\hat{\epsilon}_i - \epsilon_i^\star\|_2^2. \tag{79}$$

This is particularly important for capturing the energy spectrum relevant to ground-state electronic properties. We identify the occupied orbitals by selecting the $\lfloor N/2 \rfloor$ lowest eigenvalues, where $N$ is the number of electrons in the system.

**Orbital coefficient similarity score ($\mathcal{S}_c$).** To compare the predicted and ground-truth molecular orbital coefficient matrices $\hat{\mathbf{C}}$ and $\mathbf{C}^\star$, we use a cosine similarity-based score averaged over all columns:

$$\mathcal{S}_C(\hat{\mathbf{C}}, \mathbf{C}^\star) = \left\langle \hat{\mathbf{C}}, \mathbf{C}^\star \right\rangle = \frac{\sum_i \mathbf{C}_i \mathbf{C}_i^\star}{\|\hat{\mathbf{C}}\| \|\mathbf{C}^\star\|}, \tag{80}$$

where $\mathbf{C}_i$ is the $i$-th column vector of the coefficient matrix and $\langle \cdot, \cdot \rangle$ denotes the inner product. This metric evaluates how well the predicted orbitals align with the ground truth.

**HOMO, LUMO, and energy gap MAE ($\epsilon_{\mathbf{HOMO}}$, $\epsilon_{\mathbf{LUMO}}$, $\epsilon_\Delta$).** These metrics quantify the predictive accuracy of specific frontier orbital energies, which are critical for determining electronic properties such as reactivity and charge transport. The highest occupied molecular orbital (HOMO) and lowest unoccupied molecular orbital (LUMO) energies, along with their difference (the HOMO–LUMO gap), are particularly sensitive to model generalization, especially in challenging settings like QH9, where predictions are evaluated on unseen molecular structures or conformations. These metrics are computed as follows:

$$\epsilon_{\mathrm{HOMO}} = \left| \hat{\epsilon}_{\mathrm{HOMO}} - \epsilon_{\mathrm{HOMO}}^\star \right|, \tag{81}$$

$$\epsilon_{\mathrm{LUMO}} = \left| \hat{\epsilon}_{\mathrm{LUMO}} - \epsilon_{\mathrm{LUMO}}^\star \right|, \tag{82}$$

$$\epsilon_\Delta = \left| (\hat{\epsilon}_{\mathrm{LUMO}} - \hat{\epsilon}_{\mathrm{HOMO}}) - (\epsilon_{\mathrm{LUMO}}^\star - \epsilon_{\mathrm{HOMO}}^\star) \right|. \tag{83}$$

Here, $\epsilon_{\mathrm{HOMO}}$ corresponds to the $\lfloor N/2 \rfloor$-th lowest eigenvalue, and $\epsilon_{\mathrm{LUMO}}$ to the $(\lfloor N/2 \rfloor + 1)$-th, where $N$ is the number of electrons in the system.

**SCF Iter Ratio.** This metric measures the reduction in the number of SCF iterations required when using a predicted Hamiltonian as the initial guess, relative to a standard reference initialization (e.g., from a minimal basis or atomic superposition guess). It is defined as

$$\mathrm{SCF\ Iter\ Ratio} = \frac{\mathtt{Iter}_{\mathrm{pred}}}{\mathtt{Iter}_{\mathrm{ref}}}, \tag{84}$$

where $\mathtt{Iter}_{\mathrm{pred}}$ is the number of SCF iterations with the predicted initialization, and $\mathtt{Iter}_{\mathrm{ref}}$ is the number of iterations under the reference setup. In our setting, we used the reference value, which is measured from the conventional DFT metric to compare the acceleration of our method.

**SCF T Ratio.** This represents the reduction in total wall-clock time spent during SCF convergence using the predicted Hamiltonian. It reflects actual runtime savings in quantum chemical simulations:

$$\text{SCF T Ratio} = \frac{T_{\text{SCF-pred}}}{T_{\text{SCF-ref}}}, \tag{85}$$

where $T_{\text{SCF-pred}}$ and $T_{\text{SCF-ref}}$ denote the SCF runtimes under predicted and reference initializations, respectively.

**Inf T Ratio.** This metric captures the inference overhead of the predictive model itself. It is the fraction of time required to generate the predicted Hamiltonian relative to the baseline SCF time:

$$\text{Inf T Ratio} = \frac{T_{\text{inference}}}{T_{\text{SCF-ref}}}, \tag{86}$$

where $T_{\text{inference}}$ is the runtime of the Hamiltonian prediction model.

**Total T Ratio.** This quantifies the net cost of using the predictive model, combining both inference and SCF convergence times. It provides a holistic view of overall computational efficiency:

$$\text{Total T Ratio} = \frac{T_{\text{inference}} + T_{\text{SCF-pred}}}{T_{\text{SCF-ref}}}. \tag{87}$$

**Note.** All energy values are reported in units of Hartree ($1E_h = 27.211eV$), and similarity scores are unitless with values in $[0, 1]$, where higher is better. While the SCF `Iter` Ratio is a deterministic metric reflecting algorithmic convergence behavior, the wall-clock timing ratios (SCF `T`, Inf `T`, and Total `T`) are subject to variability due to system-level factors. In our experiments, all timing evaluations were conducted on A100-80GB GPUs. However, due to resource allocation via SLURM and shared CPU environments, precise time measurements were not always reproducible. As such, reported time ratios should be interpreted as indicative trends rather than absolute benchmarks.

### E.3   Experimental setup

**Environment.** Experiments were conducted using a single GPU per model. For the MD17 dataset, we used NVIDIA RTX 3090 and A5000 GPUs, while for the QH9 dataset, experiments were performed on NVIDIA A100 80GB GPUs. Our implementation is based on PyTorch 2.1.2 and PyG 2.3.0, both compiled with CUDA 12.1. Detailed package versions [62–64] and environment specifications will be released upon publication for full reproducibility.

Training was performed on a SLURM-managed cluster, where slight variability in runtime was observed due to system-wide GPU and CPU resource contention. Random seeds were fixed where possible to ensure training stability; however, minor non-determinism remains due to the inherent variability of distributed computing environments.

Training QHFlow took approximately 2–4 days on MD17 and 7–10 days on QH9, which is slightly longer than QHNet due to the added complexity of incorporating the current Hamiltonian and orbital information as inputs. Finetuning on QH9 required an additional 3–4 days. While the total training time could be further reduced with optimized infrastructure, our experiments were conducted under a SLURM-managed environment with periodic reinitialization every 72 hours. Importantly, QHFlow is a model-agnostic framework; with more scalable architectures, training efficiency can be significantly improved.

**Hyperparameters.** For fair comparison, we adopted the same hyperparameters used by the baseline model QHNet [23] whenever possible. Our QHFlow shares the majority of its architecture and training settings with QHNet to ensure that performance improvements are attributable to the proposed flow matching design rather than hyperparameter tuning. A summary of the key hyperparameters used across different datasets is provided in Table 7.

For evaluating DFT acceleration performance, we used the PySCF [62] framework with the B3LYP exchange–correlation functional. The numerical integration grid level was set to 3, and the maximum number of SCF cycles was 50. All other parameters followed the default settings of the PySCF RKS implementation.

Table 7: Training hyperparameters of QHFlow used across datasets.

| Hyperparameter | Description | QH9 | MD17 (Water) | MD17 (Others) |
|---|---|---|---|---|
| Learning Rate | Initial learning rate | 5e-4 | 5e-4 | 1e-3 |
| Minimum Learning Rate | Minimum learning rate | 1e-7 | 1e-9 | 1e-9 |
| Batch Size | Number of molecules per batch | 32 | 10 | 5 |
| Scheduler | Learning rate scheduler | Polynomial | Polynomial | Polynomial |
| LR Warmup Steps | Warmup steps to linearly increase learning rate | 1,000 | 1,000 | 1,000 |
| Max Steps | Maximum number of training steps | 260,000 | 200,000 | 200,000 |
| Fine-tuning LR | Initial learning rate of fine-tuning stage | 1e-5 | - | - |
| Fine-tuning Minimum LR | Minimum learning rate of fine-tuning stage | 1e-7 | - | - |
| Finetuning Steps | Maximum number of Finetuning steps | 60,000 | - | - |
| Prior Distribution | Prior distribution for flow matching | GOE / TE | GOE | GOE |
| Using $H_t$ Block | Use time-dependent Hamiltonian $H_t$ as input | True | True | True |
| Using $S$ Block | Use overlap matrix $S$ as input | True | False | False |
| Model Order | Maximum degree of spherical harmonics | 4 | 4 | 4 |
| Embedding Dimension | Node feature embedding dimension | 128 | 128 | 128 |
| Bottle Hidden Size | Hidden size of bottleneck layer | 32 | 32 | 32 |
| Number of GNN Layers | Number of graph neural network layers | 5 | 5 | 5 |
| Max Radius | Cutoff radius for neighbor search | 15 | 15 | 15 |
| Sphere Channels | Number of channels in spherical basis | 128 | 128 | 128 |
| Edge Channels | Number of channels for edge features | 32 | 32 | 32 |

# F  Additional experimental results and limitations

## F.1  Ablation study of the finetuning objective

We conduct additional experiments to evaluate various fine-tuning strategies to improve the Hamiltonian prediction performance of QHFlow after standard flow matching pretraining. In this section, we describe each strategy in detail and provide empirical observations.

For the WALoss implementation, we follow the coefficient settings proposed in WANet [25] ($\lambda_{WA} = 2.0$). During fine-tuning, we train for an additional 60,000 steps starting from the pretrained model with TE prior, using an initial learning rate of $1 \times 10^{-5}$ and a polynomial learning rate scheduler.

**Full Flow Matching with WA Loss training from scratch (WA-Full).** In this strategy, we train the model from scratch by jointly optimizing the flow matching objective and WALoss for 260,000 steps, matching the default training steps as done in WANet. However, this approach significantly degraded the Hamiltonian error. We hypothesize that imposing strong spectral constraints too early disrupts the learning of the global Hamiltonian structure, leading to unstable convergence and poor generalization.

**WA Loss finetuning (WA-FT).** After completing standard flow matching pre-training, we fine-tune the model solely using WALoss, which encourages alignment between the predicted Hamiltonian's eigenstructure and the ground-truth SCF solutions. This approach improves orbital energy prediction and SCF convergence compared to pure flow matching training, confirming the value of adding spectral supervision during fine-tuning.

**Summary.** Table 8 summarizes the quantitative results of all fine-tuning strategies. Here, w/o-FT refers to QHFlow trained solely with the flow matching objective using the TE prior. Among the approaches, WA-FT achieves the best balance between Hamiltonian prediction accuracy and practical DFT acceleration, demonstrating the effectiveness of spectral alignment in the fine-tuning stage.

## F.2  Full results of DFT acceleration via SCF initialization

For completeness, we include in Figure 5 the figure with the inset from Figure 3. As shown in Figure 5, the inference time of QHFLOW is longer than that of QHNet because QHFLOW performs multi-step inference for ODE integration. However, this additional cost is negligible compared to the overall DFT computation pipeline, as reflected in the *SCF T ratio*.

## F.3  Full results of ablation study

For completeness, we provide the full ablation study results in Table 9, and Table 10, including additional metrics beyond those presented in the main text. While the main manuscript focused primarily on Hamiltonian prediction error ($H$), occupied orbital energy error ($\epsilon_{occ}$), and electronic density accuracy ($\mathcal{S}_c$) for clarity, here we also report the LUMO ($\epsilon_{LUMO}$), HOMO ($\epsilon_{HOMO}$), and HOMO-LUMO gap ($\epsilon_\Delta$) energy errors.

Overall, the trends observed in the full table are consistent with those discussed in the main text. In particular, the improvements in Hamiltonian accuracy and density prediction are accompanied by consistent reductions in orbital energy errors ($\epsilon_{LUMO}$, $\epsilon_{HOMO}$) and gap errors ($\epsilon_\Delta$), further demonstrating the broad effectiveness of our proposed methods. These results reinforce the conclusion that flow matching enhance both Hamiltonian structure prediction and downstream electronic properties.

Also, to better visualize the role of the prior distributions, we provide interpolation trajectories from the GOE and TE prior toward target Hamiltonians. Figure 6 shows example interpolations for both settings, highlighting how the initial distribution affects the learned flow dynamics. As seen, the TE prior results in smoother and more structured trajectories compared to the GOE prior, which starts from unstructured noise.

## F.4  Ablation studies on model design

In this section, we present additional ablation studies on the architectural design choices of QHFLOW, as summarized in Table 11, to clarify their individual roles and contributions to the overall performance.

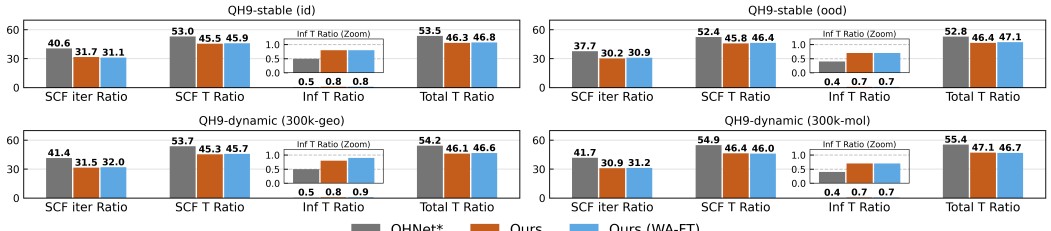

Figure 5: **DFT acceleration performance on 300 samples from the QH9 dataset.** All metrics are reported as percentages relative to conventional DFT (initialized with `minao`), which serves as the 100% baseline. The *SCF Iter Ratio* measures the ratio of SCF iterations required, while *Inf T Ratio*, *SCF T Ratio*, and *Total T Ratio* measure time.

**Regression.** QHFLOW (Regression) is a baseline that directly predicts the Hamiltonian correction matrix ($\Delta \mathbf{H} = \mathbf{H}^\star - \mathbf{H}^{(0)}$) using the same objective described above, without incorporating any physical priors or structural constraints. This variant isolates the effect of the flow-matching mechanism itself and evaluates how well the model performs under a purely regression-based setting. Although this approach achieves improved Hamiltonian prediction compared to QHNet, its overall performance remains inferior to the flow-matching version.

**Overlap matrix S.** We further examine the impact of incorporating the overlap matrix embedding, which encodes pairwise correlations among atomic orbitals. Comparing models with and without **S** embedding shows that leveraging overlap information significantly enhances model stability and accuracy across both Hamiltonian prediction and SCF convergence.

### F.5 Analysis of error distribution of the predictions

To better understand the properties of QHFLOW, we move beyond single–number summaries and visualise the entire error distribution for six key quantities Hamiltonian MAE ($H$), occupied–orbital MAE ($\epsilon_{\text{occ}}$), $\epsilon_{\text{HOMO}}$, $\epsilon_{\text{LUMO}}$, $\epsilon_\Delta$, and $\mathcal{S}_c$ on all four QH9 splits (`id`, `ood`, `geo`, `mol`). Figure 7 shows log-scale violin plots for QHNet* (grey), QHFLOW (orange), and its WA-finetuned variant (cyan). (only the similarity score is kept on a linear axis.) Each violin displays the full empirical distribution; the thick dashed line marks the median, and the two thinner lines divide the inter-quartile ranges.

Across every split and metric, the QHFLOW violins are both *lower* and *narrower*. Median Hamiltonian error is reduced by roughly one order of magnitude on `id` and `geo`, and by a factor of 3–5 on the tougher `ood` and `mol` sets. In addition, the long error tails of QHNet* largely vanish, demonstrating that our flow objective mitigates the occasional catastrophic failure modes that plague point-wise regression models.

Figure 8 plots $\log H$ against $\log \epsilon_{\text{occ}}$ for every test geometry. QHFLOW occupies the bottom-left corner of each panel, while QHNet* points fan out towards larger joint errors, especially in `ood` and `mol`, where chemical diversity is highest. These results show the QHFLOW superior performance on both the geometric and unseen molecules.

Finally, Figure 9 compares Hamiltonian error versus atom count. QHFLOW remains flat up to the largest molecules in QH9 (29atoms). This suggests that modeling a *distribution* over Hamiltonians confers a form of size transferability.

Table 8: **Ablation experimental results on QH9 dataset Finetuning.** Results shown in **bold** denote the best result in each column, whereas those that are underlined indicate the second best.

| Dataset | Type | $H\ [\mu E_h]\downarrow$ | $\epsilon_{\text{occ}}\ [\mu E_h]\downarrow$ | $\mathcal{S}_c\ [\%]\uparrow$ | $\epsilon_{\text{LUMO}}\ [\mu E_h]\downarrow$ | $\epsilon_{\text{HOMO}}\ [\mu E_h]\downarrow$ | $\epsilon_\Delta\ [\mu E_h]\downarrow$ |
|---|---|---|---|---|---|---|---|
| QH9-stable (id) | WA-Full | 62.69 | 153.82 | 98.96 | 240.07 | 117.26 | 251.81 |
| | w/o-FT | $\mathbf{22.95}_{\pm 0.001}$ | $119.67_{\pm 0.211}$ | $99.51_{\pm 0.001}$ | $437.96_{\pm 7.452}$ | $179.48_{\pm 0.098}$ | $553.87_{\pm 7.454}$ |
| | WA-FT | $23.85_{\pm 0.003}$ | $\mathbf{101.92}_{\pm 0.279}$ | $\mathbf{99.56}_{\pm 0.002}$ | $\mathbf{187.48}_{\pm 6.434}$ | $\mathbf{92.22}_{\pm 0.234}$ | $\mathbf{206.15}_{\pm 6.197}$ |
| QH9-stable (ood) | WA-Full | 57.11 | 110.34 | 98.12 | 416.08 | 113.55 | 387.83 |
| | w/o-FT | $\mathbf{20.01}_{\pm 0.001}$ | $84.54_{\pm 0.007}$ | $99.04_{\pm 0.003}$ | $321.20_{\pm 1.497}$ | $130.74_{\pm 0.043}$ | $395.83_{\pm 1.510}$ |
| | WA-FT | $20.55_{\pm 0.002}$ | $\mathbf{72.64}_{\pm 0.018}$ | $\mathbf{99.16}_{\pm 0.006}$ | $171.24_{\pm 0.273}$ | $\mathbf{77.96}_{\pm 0.095}$ | $\mathbf{179.57}_{\pm 0.271}$ |
| QH9-dynamic (300k-geo) | WA-Full | 75.38 | 108.96 | 98.77 | 231.63 | 114.14 | 219.20 |
| | w/o-FT | $\mathbf{25.94}_{\pm 0.001}$ | $103.11_{\pm 0.031}$ | $99.59_{\pm 0.001}$ | $425.18_{\pm 1.119}$ | $175.18_{\pm 0.255}$ | $547.33_{\pm 1.168}$ |
| | WA-FT | $27.12_{\pm 0.002}$ | $\mathbf{89.03}_{\pm 0.213}$ | $\mathbf{99.65}_{\pm 0.001}$ | $136.63_{\pm 4.661}$ | $\mathbf{84.17}_{\pm 0.211}$ | $\mathbf{154.68}_{\pm 4.449}$ |
| QH9-dynamic (300k-mol) | WA-Full | 103.47 | 692.17 | 97.02 | 1046.21 | 760.85 | 1408.36 |
| | w/o-FT | $\mathbf{45.91}_{\pm 0.001}$ | $442.56_{\pm 0.171}$ | $98.65_{\pm 0.001}$ | $1344.68_{\pm 2.338}$ | $479.71_{\pm 0.150}$ | $1605.03_{\pm 2.286}$ |
| | WA-FT | $46.60_{\pm 0.003}$ | $\mathbf{424.75}_{\pm 0.324}$ | $\mathbf{98.74}_{\pm 0.001}$ | $\mathbf{912.10}_{\pm 2.941}$ | $403.51_{\pm 1.861}$ | $\mathbf{1047.88}_{\pm 2.683}$ |

Table 9: **Full experimental results on QH9 dataset along the initial distribution.** Results shown in **bold** denote the best result in each column.

| Dataset | Prior | $H\downarrow$ $[10^{-6}E_h]$ | $\epsilon_{\text{occ}}\downarrow$ $[10^{-6}E_h]$ | $\mathcal{S}_c\uparrow$ $[10^{-2}]$ | $\epsilon_{\text{LUMO}}\downarrow$ $[10^{-6}E_h]$ | $\epsilon_{\text{HOMO}}\downarrow$ $[10^{-6}E_h]$ | $\epsilon_\Delta\downarrow$ $[10^{-6}E_h]$ |
|---|---|---|---|---|---|---|---|
| QH9-stable (id) | GOE | $25.93_{\pm 0.001}$ | $154.65_{\pm 1.097}$ | $99.39_{\pm 0.001}$ | $638.03_{\pm 7.744}$ | $220.49_{\pm 0.192}$ | $764.46_{\pm 7.635}$ |
| | TE | $\mathbf{22.95}_{\pm 0.001}$ | $\mathbf{119.61}_{\pm 0.211}$ | $\mathbf{99.51}_{\pm 0.001}$ | $\mathbf{437.96}_{\pm 7.452}$ | $\mathbf{179.48}_{\pm 0.098}$ | $\mathbf{553.87}_{\pm 7.454}$ |
| QH9-stable (ood) | GOE | $21.93_{\pm 0.001}$ | $87.32_{\pm 0.012}$ | $98.95_{\pm 0.002}$ | $382.09_{\pm 13.87}$ | $\mathbf{120.16}_{\pm 0.070}$ | $432.93_{\pm 13.84}$ |
| | TE | $\mathbf{20.01}_{\pm 0.001}$ | $\mathbf{84.54}_{\pm 0.007}$ | $\mathbf{99.04}_{\pm 0.003}$ | $\mathbf{321.20}_{\pm 1.497}$ | $130.74_{\pm 0.043}$ | $\mathbf{395.83}_{\pm 1.510}$ |
| QH9-dynamic (300k-geo) | GOE | $29.39_{\pm 0.001}$ | $122.14_{\pm 0.050}$ | $99.49_{\pm 0.001}$ | $\mathbf{618.75}_{\pm 3.089}$ | $215.41_{\pm 0.569}$ | $\mathbf{756.45}_{\pm 3.371}$ |
| | TE | $\mathbf{25.94}_{\pm 0.001}$ | $\mathbf{103.11}_{\pm 0.031}$ | $\mathbf{99.59}_{\pm 0.001}$ | $425.18_{\pm 1.119}$ | $\mathbf{175.18}_{\pm 0.255}$ | $547.33_{\pm 1.168}$ |
| QH9-dynamic (300k-mol) | GOE | $46.78_{\pm 0.001}$ | $\mathbf{419.68}_{\pm 0.001}$ | $\mathbf{98.65}_{\pm 0.001}$ | $1409.07_{\pm 3.759}$ | $\mathbf{478.12}_{\pm 0.214}$ | $1718.80_{\pm 3.907}$ |
| | TE | $\mathbf{45.91}_{\pm 0.001}$ | $442.56_{\pm 0.171}$ | $\mathbf{98.65}_{\pm 0.001}$ | $\mathbf{1344.68}_{\pm 2.338}$ | $479.71_{\pm 0.150}$ | $\mathbf{1605.03}_{\pm 2.286}$ |

Table 10: **Full predictive variance results on QH9 dataset.** WA-FT implies the finetune the model with WA loss. Metrics for Ours and Ours (WA-FT) are means±std over five random seeds. Results shown in **bold** denote the best result in each column, whereas those that are underlined indicate the second best.

| Dataset | Model | $H\ [\mu E_h]\downarrow$ | $\epsilon_{\text{occ}}\ [\mu E_h]\downarrow$ | $\mathcal{S}_c\ [\%]\uparrow$ | $\epsilon_{\text{LUMO}}\ [\mu E_h]\downarrow$ | $\epsilon_{\text{HOMO}}\ [\mu E_h]\downarrow$ | $\epsilon_\Delta\ [\mu E_h]\downarrow$ |
|---|---|---|---|---|---|---|---|
| QH9-stable (id) | QHNet* | 77.72 | 963.45 | 94.80 | 18257.34 | 1546.27 | 17822.62 |
| | WANet | 80.00 | 833.62 | 96.86 | - | - | - |
| | SPHNet | 45.48 | 334.28 | 97.75 | - | - | - |
| | **Ours** | $\mathbf{22.95}_{\pm 0.001}$ | $119.67_{\pm 0.211}$ | $99.51_{\pm 0.001}$ | $437.96_{\pm 7.452}$ | $179.48_{\pm 0.098}$ | $553.87_{\pm 7.454}$ |
| | **Ours (WA-FT)** | $23.85_{\pm 0.003}$ | $\mathbf{101.92}_{\pm 0.279}$ | $\mathbf{99.56}_{\pm 0.002}$ | $\mathbf{187.48}_{\pm 6.434}$ | $\mathbf{92.22}_{\pm 0.234}$ | $\mathbf{206.15}_{\pm 6.197}$ |
| QH9-stable (ood) | QHNet* | 69.69 | 884.97 | 93.01 | 25848.83 | 1045.99 | 25370.10 |
| | SPHNet | 43.33 | 186.40 | 98.16 | - | - | - |
| | **Ours** | $\mathbf{20.01}_{\pm 0.001}$ | $84.54_{\pm 0.007}$ | $99.04_{\pm 0.003}$ | $321.20_{\pm 1.497}$ | $130.74_{\pm 0.043}$ | $395.83_{\pm 1.510}$ |
| | **Ours (WA-FT)** | $20.55_{\pm 0.002}$ | $\mathbf{72.64}_{\pm 0.018}$ | $\mathbf{99.16}_{\pm 0.006}$ | $171.24_{\pm 0.273}$ | $77.96_{\pm 0.095}$ | $179.57_{\pm 0.271}$ |
| QH9-dynamic (300k-geo) | QHNet* | 88.36 | 1170.50 | 93.65 | 23269.41 | 2040.06 | 22407.96 |
| | WANet | 74.74 | 416.57 | 99.68 | - | - | - |
| | SPHNet | 52.18 | 100.88 | 99.12 | - | - | - |
| | **Ours** | $\mathbf{25.94}_{\pm 0.001}$ | $103.11_{\pm 0.031}$ | $99.59_{\pm 0.001}$ | $425.18_{\pm 1.119}$ | $175.18_{\pm 0.255}$ | $547.33_{\pm 1.168}$ |
| | **Ours (WA-FT)** | $27.12_{\pm 0.002}$ | $\mathbf{89.03}_{\pm 0.213}$ | $\mathbf{99.65}_{\pm 0.001}$ | $136.63_{\pm 4.661}$ | $84.17_{\pm 0.211}$ | $154.68_{\pm 4.449}$ |
| QH9-dynamic (300k-mol) | QHNet* | 121.39 | 5554.36 | 86.02 | 53505.09 | 4352.76 | 50424.86 |
| | SPHNet | 108.19 | 1724.10 | 91.49 | - | - | - |
| | **Ours** | $\mathbf{45.91}_{\pm 0.001}$ | $442.56_{\pm 0.171}$ | $98.65_{\pm 0.001}$ | $1344.68_{\pm 2.338}$ | $479.71_{\pm 0.150}$ | $1605.03_{\pm 2.286}$ |
| | **Ours (WA-FT)** | $46.60_{\pm 0.003}$ | $424.75_{\pm 0.324}$ | $98.74_{\pm 0.001}$ | $912.10_{\pm 2.941}$ | $403.51_{\pm 1.861}$ | $1047.88_{\pm 2.683}$ |

Table 11: **Ablation study on the model design of QHFLOW.** The models are trained on the QH9-Stable (id) split. **Bold** indicate the best.

| Model | $H\downarrow [\mu E_h]$ | $\epsilon_{\text{occ}}\downarrow [\mu E_h]$ | $\mathcal{S}_c\uparrow [\%]$ | $\epsilon_{\text{LUMO}}\downarrow [\mu E_h]$ | $\epsilon_{\text{HOMO}}\downarrow [\mu E_h]$ | $\epsilon_\Delta\downarrow [\mu E_h]$ |
|---|---|---|---|---|---|---|
| QHNet* | 77.72 | 963.45 | 94.80 | 18257.34 | 1546.27 | 17822.62 |
| QHFLOW (Regression) | 35.85 | 213.45 | 98.89 | 1548.64 | 330.77 | 1783.99 |
| QHFLOW (W/o $\mathbf{S}$ embed) | 25.24 | 159.41 | 99.38 | 734.40 | 239.18 | 851.33 |
| QHFLOW (W/ $\mathbf{S}$ embed) | **22.95** | **119.67** | **99.51** | **437.96** | **179.48** | **553.87** |

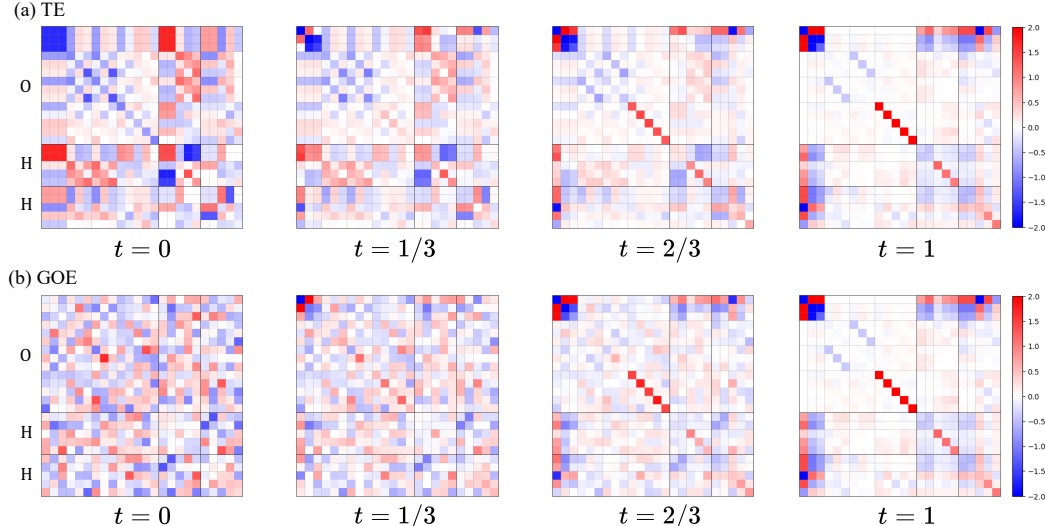

Figure 6: Visualization of Hamiltonian interpolation trajectories from different prior distributions of $H_2O$: (a) tensor expansion-based (TE) prior and (b) Gaussian orthogonal ensemble (GOE) prior. Color intensity indicates the magnitude of Hamiltonian elements across flow time $t$.

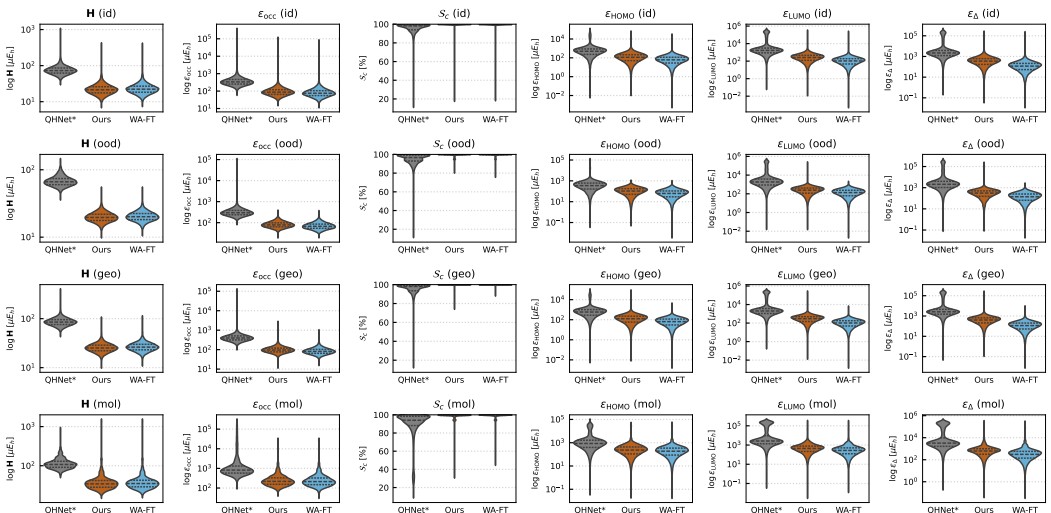

Figure 7: Violin plots compare QHNet* (gray), QHFLOW(orange), and its weighted-alignment fine-tune (WA-FT, cyan) on six metrics: Hamiltonian MAE ($\mathbf{H}$), occupied-orbital MAE ($\epsilon_{occ}$), $\epsilon_{HOMO}$, $\epsilon_{LUMO}$, $\epsilon_\Delta(\Delta)$, and coefficient similarity ($\mathcal{S}_c$). Results are shown for the four evaluation splits — id, ood, geo, mol. All axes are logarithmic except $\mathcal{S}_c$. Thick dashed lines mark the median; thin dashed lines indicate the quartiles. Across every split the QHFLOWviolins are both lower and narrower than those of QHNet*, and WA-FT delivers an additional (though smaller) improvement on the energy-related metrics.

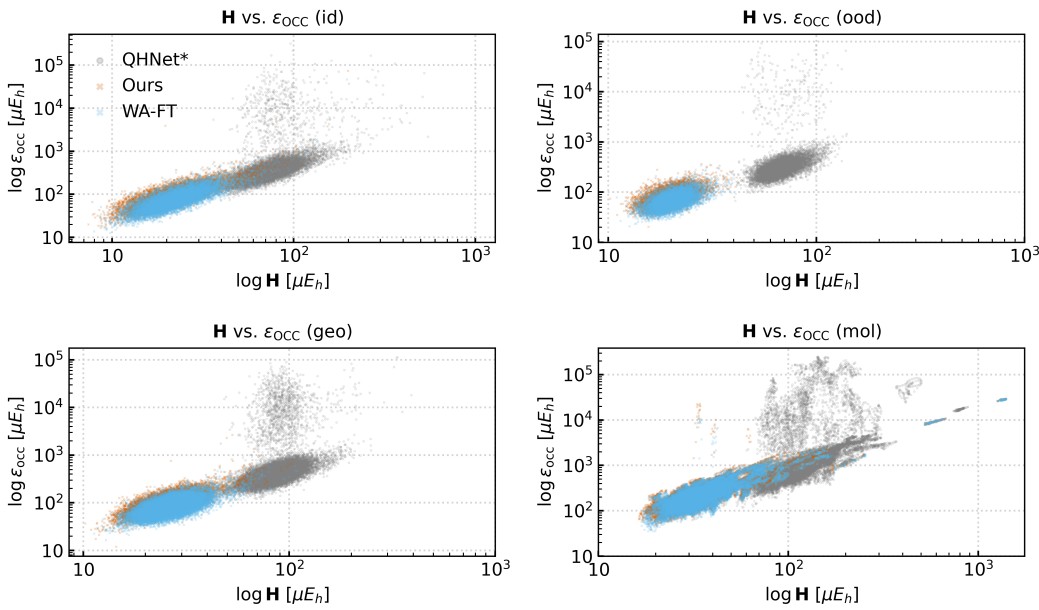

Figure 8: Scatter plots of $\log \mathbf{H}$ versus $\log \epsilon_{\text{occ}}$ for each QH9 split. Points for QHNet* (gray) extend to the upper right, revealing frequent large joint errors. QHFLOW (orange) and WA-FT (cyan) cluster tightly in the lower-left region, indicating that the flow-matching model achieves simultaneously low Hamiltonian and occupied-orbital errors, even on the challenging ood and mol splits.

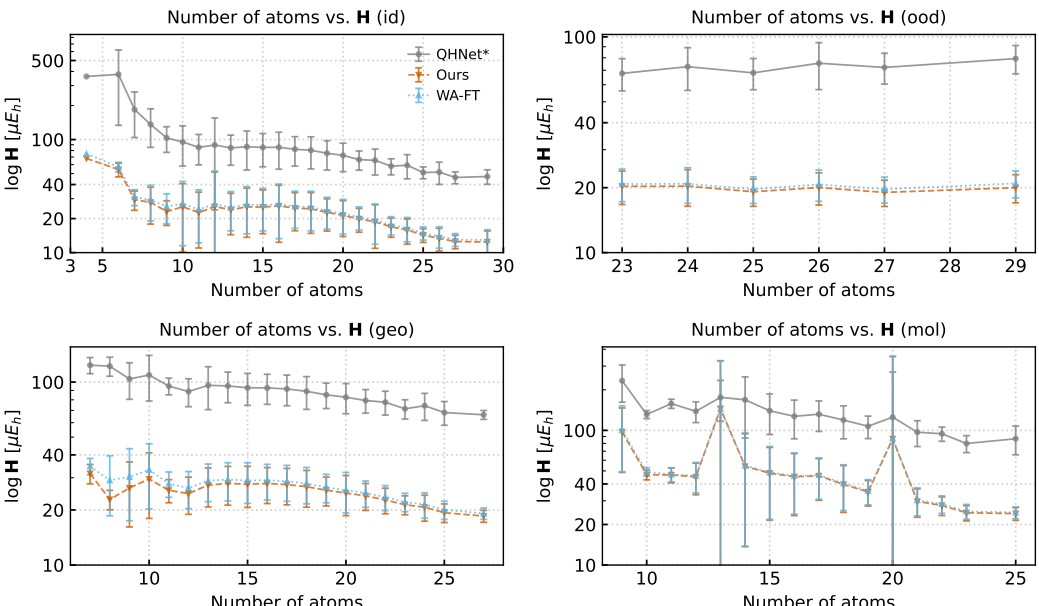

Figure 9: Mean log Hamiltonian error (markers) and one-standard-deviation bands (vertical bars) versus atom count for the four QH9 splits. QHNet* (gray) errors rise with molecular size in ood, whereas QHFLOW (orange) and WA-FT (cyan) remain nearly flat up to 29 atoms. This suggests that modelling the full Hamiltonian distribution with symmetry-aware priors confers robust transferability to larger, more complex molecules.

