# OpenReview forum: "High-order Equivariant Flow Matching for Density Functional Theory Hamiltonian Prediction"
_NeurIPS.cc/2025/Conference — NeurIPS 2025 spotlight_

### Official Review · Reviewer_UyjR · 2025-06-02

**Clarity:** 4
**Significance:** 3
**Originality:** 3
**Rating:** 5
**Confidence:** 3

**Summary:**

Deep learning approaches for prediction of the Hamiltonian matrix offer a way to improve the efficiency of density functional theory (DFT) calculations by forgoing or reducing the number of self-consistent field (SCF) steps needed. The authors propose a SE(3)-equivariant flow matching model for the prediction of this Hamiltonian. This is achieved through the introduction of a symmetry-aware prior distribution and orbital energy finetuning procedure. On several datasets, this method demonstrates improved performance in Hamiltonian error metrics, and reduction in SCF steps compared to prior work that predicts the Hamiltonian through regression.

**Questions:**

* How much of the performance improvement over QHNet is due to architectural changes (see *Weaknesses*) vs. the introduction of flow matching? It would be helpful to expand the ablation studies in section 5.4 to include either a QHNet baseline with the architectural improvements and energy alignment fine-tuning, or a QHFlow implementation without the architectural improvements, so we can better understand the source of the improved performance.
* Fig. 3: Given QHFlow is evaluated with "3-step ODE-based sampling" wouldn't we expect the inference time ratio for QHFLow to be roughly 3 times that of QHNet? While the bar colors of "Inf T Ratio" can be inferred by order, it would be helpful to modify this figure so that this is more obvious.
* What datapoints are the "71%" M17 and "53%" QH9 Hamiltonian derived from? The authors should be more precise with these statements in both the abstract and results section. Measuring the mean % improvement over the most recent state-of-the-art method would make the most sense to me.

**Ethical Concerns:**

["NO or VERY MINOR ethics concerns only"]

**Final Justification:**

The authors propose a SE(3)-equivariant flow matching model for the prediction of the molecular Hamiltonian from density functional theory (DFT). While there are existing methods for predicting this quantity through regression, this work is the first to demonstrate the use of a generative framework for the task. With this flow matching framework, they demonstrate an improvement over prior methods both in terms of prediction error, but also self-consistent field (SCF) step reduction in simulation on several benchmark tasks. They demonstrate through ablation that this is due to both architectural improvements and the proposed flow-matching method.

* *Why I am not rating it higher:* While the work demonstrates clear improvements in the Hamiltonian prediction task, I would consider it more of an incremental improvement in modeling than a "groundbreaking impact". Furthermore the methods and analysis introduced likely have limited applicability in other areas of AI.
* *Why I am not rating it lower:* The use of flow matching in the prediction of Hamiltonians is a novel and interesting contribution, and the authors provide adequate evaluation and analysis.

**Limitations:**

The authors do not include any limitations of their work. One notable limitation in this line of research is that Hamiltonian prediction error is not always well correlated with SCF step reduction. I.e. a model that has a very low $\mathbf{H}$ MAE might not necessarily improve DFT runtimes, which could suggest a different task if SCF step reduction is the main goal. This does not seem to be the case in this work (at least in aggregate) but could be worth discussing (along with other things).

**Quality:**

3

**Strengths And Weaknesses:**

### Strengths

* *Clarity*: the paper is generally well written and easy to understand.
* *Scope*: the problem is well defined, and the work includes two distinct benchmark tasks.
* *Originality*: while SE(3) equivariant flow matching models have been proposed in the past, this work is the first to apply them to the task of Hamiltonian prediction, which has generally been done through direct regression. This is non-trivial due to the higher-order nature of the Hamiltonian matrix.
* *Significance*: the proposed method offers an improved accuracy of both Hamiltonian prediction and SCF step reduction compared to prior work.

### Weaknesses
 * The authors make several modifications to the QHNet architecture in their method, including the additional input of the overlap matrix $\mathbf{S}$, attention-based message-passing layers, and a channel mixing module. Furthermore, the introduced energy alignment fine-tuning could also be applied to the regression-based model. Given the focus of the work is on the proposed flow-matching method, it is unclear if the performance gains are due to the generative task or improvements in model architecture.

---

> ### Author Rebuttal · Authors · 2025-07-30
>
> Dear Reviewer UyjR,
>
> Thank you for your thoughtful and encouraging review. We greatly appreciate your recognition of the clarity of our writing, the originality of applying SE(3)-equivariant flow matching to Hamiltonian prediction, and the significance of our results in both Hamiltonian accuracy and SCF step reduction. Below, we address your comments in detail.
>
> ---
>
> **W1, Q1: It is unclear if the performance gains are due to the generative task or improvements in model architecture.**
>
> We appreciate the reviewer’s question regarding the source of performance improvement. To address this concern, we conducted an ablation study comparing our flow-based framework to a regression-based baseline under identical architectural conditions:
>
> | Model (QH9-id) | H mae $\downarrow$ [$\mu E_h$] | $\epsilon_{occ}$ $\downarrow$ [$\mu E_h$] | $\mathcal S_c$ $\uparrow$ [%]    |   LUMO $\downarrow$ [$\mu E_h$] | HOMO $\downarrow$ [$\mu E_h$] | GAP $\downarrow$ [$\mu E_h$] |
> | -------- | --------: | --------: |--------: | --------: | --------: |--------: |
> | QHNet     |  77.72     | 963.45     | 94.80  |   18257.34 |  1546.27 |  17822.62 |
> | QHFlow (*Regression*)    | 35.85     | 213.45     | 98.89  |   1548.64 |  330.77 |  1738.99 |
> | QHFlow (GOE)    | 25.93     | 154.65    | 99.39 | 638.03  |  220.49  |       764.46 |
> | QHFlow (TE)   | **22.95**     | 119.67     | 99.51   |      437.96   | 179.48   | 553.87  |
> | QHFlow (TE, WA-FT)    | 23.85     | **101.92**     | **99.56**   |  **187.48**       |   **92.22**    |  **206.15**    |
>
> These results demonstrate that both architectural modifications and the adoption of flow matching contribute meaningfully to our performance gains. Notably, switching from regression to flow-based training yields consistent improvements across all metrics, even under the same model architecture. Furthermore, the choice of prior distribution (TE vs. GOE) and the use of a fine-tuning strategy further highlight the trade-off between Hamiltonian accuracy and energy prediction.
>
> We will include the complete ablation results in the final version and publicly release our codebase to support further research in this area.
>
> ---
> **Q2-1: (Figure 3) Given QHFlow is evaluated with "3-step ODE-based sampling," wouldn't we expect the inference time ratio for QHFlow to be roughly 3 times that of QHNet?**
>
> This is due to additional latency that is constant across QHFlow and QHNet. The latency is caused by system-level overheads such as data movement and Python runtime costs. Once we make the (unrealistic) assumption that the latency is not part of the computational cost, QHFlow would cost roughly three times that of QHNet.
>
> More importantly, we emphasize that the primary computational bottleneck in DFT workflows is the SCF iteration process. Since our method substantially reduces the number of SCF iterations, as shown in Figure 3, the added inference cost becomes negligible in the context of end-to-end simulation time.
>
> ---
>
> **Q2-2: While the bar colors of "Inf T Ratio" can be inferred by order, it would be helpful to modify this figure so that this is more obvious.**
>
> Thank you for the helpful suggestion. In the final version, we will revise the "Inf T ratio" part to be more visible and clear. Specifically, we will add a small inset above each “Inf T Ratio” bar group to zoom in on it and better highlight the differences.
>
> As external link sharing is not allowed, we are unable to upload the revised figure. We will ensure to update it in the final manuscript.
>
> ---
>
> **Q3: What datapoints are the "71%" M17 and "53%" QH9 Hamiltonian derived from?**
>
> Thank you for the question. We report the mean relative improvement in Hamiltonian MAE compared to the previous state-of-the-art model, as reducing Hamiltonian error is one of the primary objectives. We measure the improvement by $1-\frac{A}{B}$ where $A$ is the Hamiltonian MAE of our model and $B$ is that of the previous best model.
>
> For example, we observed the following improvements on MD17:
>
> - Water: 58% = $1 - \frac{4.93}{11.7}$
> - Ethanol: 73% = $1 - \frac{5.33}{20.09}$
> - Malondialdehyde: 82% = $1 - \frac{3.80}{20.67}$
> - Uracil: 80% = $1 - \frac{3.68}{18.65}$
> - Average: 71%
>
> And we observed the following improvements on QH9 (without WA-FT):
> - Stable-id: 50% = $1 - \frac{22.95}{45.48}$
> - Stable-ood: 54% = $1 - \frac{20.01}{43.33}$
> - Dynamic-300k-geo: 50% = $1 - \frac{25.94}{52.18}$
> - Dynamic-300k-mol: 58% = $1 - \frac{45.91}{108.19}$
> - Average: 53%
>
> We will revise the manuscript accordingly to reflect clarify how the improvements were calculated, as follows:
> > (line 15-16) QHFlow achieves state-of-the-art performance, reducing Hamiltonian error by 71% on MD17 and 53% on QH9 **compared to the previous best model**.
>
> > (line 289-290) With up to a 71% reduction in Hamiltonian prediction error **on average across the entire MD17 dataset** compared to the previous state-of-the-art model, QHFlow demonstrates strong predictive performance.
>
> > (line 297-298) As shown in Table 2, QHFLOW consistently outperforms all prior methods in all metrics up to a 53% reduction in Hamiltonian prediction error **on average over all QH9 splits.**
>
> We mark the text in bold to highlight its relevance to your concerns. They will be removed from the final manuscript.
>
> ---
>
> **Limitations: Lack of discussion on limitations; weak correlation between Hamiltonian prediction error and SCF acceleration.**
>
> Thank you for pointing this out! We originally included the discussion in Appendix F.4, but now realize that its visibility is low despite its importance. To alleviate your concern, we will move the limitation to the main text and revise the contents.
>
> Furthermore, we agree with your points on the lack of correlation between Hamiltonian prediction error and SCF acceleration. We think this is very insightful and a very interesting venue for future research!
>
> In the final manuscript, we will include the following text:
> > We did not fully verify the generalizability of our approach, e.g., extension to new architectures (WANet, SPHNet) or datasets (PubChemQH), due to the lack of resources, public codes, and public datasets. Our flow-based formulation introduces computational overhead for solving ODEs. In the future, one could consider removing the overhead with the one-step generative models [1]. Our experiments are limited to gas-phase molecules with up to 29 atoms and two commonly used XC functionals (PBE and B3LYP). The algorithms are left to be verified for more general settings, such as periodic solids or higher-accuracy functionals (e.g., hybrid or double-hybrid). We also acknowledge that our datasets consist of relatively small molecules. One could further test the scalability of our approach once new datasets on larger systems are available. **We also note that our work does not give a theoretical explanation of how Hamiltonian prediction error translates to SCF acceleration. Developing a theoretically grounded algorithm specifically for SCF acceleration would be an interesting future research.**
>
> We mark the text in bold to highlight its relevance to your concerns. They will be removed from the final manuscript.
>
> ---
>
> We hope our responses have addressed your questions and concerns. Please let us know if there is anything we may have missed or if any points require further clarification.
>
> [1] Kornilov, Nikita, et al. "Optimal flow matching: Learning straight trajectories in just one step." 2024 NIPS.

---

> > ### Comment · Reviewer_UyjR · 2025-08-01
> >
> > Thank you authors for taking the time to respond to my questions and comments. The added ablation study well demonstrates the effect of architectural changes and the proposed flow matching method in improving the prediction accuracy on the QH9 task. The additional proposed changes to the manuscript will improve the clarity as well. I have no further questions or comments at this time.

---

> > > ### Author Response · Authors · 2025-08-03
> > > **Response to Reviewer UyjR**
> > >
> > > Thank you for the positive feedback and for acknowledging the added ablation results and clarifications in the manuscript. We greatly appreciate your thoughtful questions and comments throughout the review process. Your insightful questions significantly enriched our presentation.

---

### Official Review · Reviewer_upG2 · 2025-06-17

**Clarity:** 4
**Significance:** 4
**Originality:** 2
**Rating:** 5
**Confidence:** 4

**Summary:**

The authors propose a flow-matching framework for generating Hamiltonians corresponding to ground-state solutions of DFT. Namely, the method builds on the QHNet architecture while adopting a generative approach in place of direct regression. To ensure SE(3) equivariance during flow matching, the method introduces two SE(3)-invariant priors: a Gaussian orthogonal ensemble and a tensor expansion-based prior. Additionally, the model extends their approach with a post-hoc fine-tuning step using a weighted alignment loss to improve the accuracy of orbital energy predictions. Experimental results on the MD17 and QH9 datasets demonstrate that the method outperforms existing regression-based baselines across multiple metrics.

**Questions:**

**Questions**
1. How does the method perform (only inference) on even larger systems e.g. Alanine tripeptide (42 atoms) and DHA (56 atoms) from the MD22 dataset?

2. Is there any intuition why the TE prior generally performs better than GOE prior?

3. Line 38-39: “However, these approaches rely on pointwise regression and often fall short 39 of capturing the full distributional structure and uncertainty inherent in the Hamiltonian.” Could the authors clarify what is meant by "distributional structure and uncertainty" in this context? To my understanding, there is generally a one-to-one mapping between structure and the ground-state solution, and it is unclear what uncertainty is referring to here.

**Suggestions**
1. Perhaps I missed it, but it would be helpful to explicitly state that Table 3 reports results from QHFlow using the TE prior. I had to match the numbers between tables to infer this connection.

**Typos**
1. Table 1: Malonaldehyde -> Malondialdehyde

**Ethical Concerns:**

["NO or VERY MINOR ethics concerns only"]

**Final Justification:**

I will maintain my score, as I believe it appropriately reflects the paper’s strong empirical performance and their rigorous experimental design, as detailed in my review. Although the paper achieves strong results using generative modeling and symmetry-aware priors for Hamiltonian prediction, the work does not introduce a fundamentally novel methodological innovation or conceptual insight. For that reason, I do not see a basis for increasing the score.

That said, the authors have addressed my questions and most of my concerns during the rebuttal, and I appreciate the clarity they brought to several points.

**Limitations:**

yes

**Quality:**

4

**Strengths And Weaknesses:**

**Strengths**
1. The paper is well written and clearly structured.

2. The proposed approach provides SOTA results by leveraging flow matching with equivariance and tailored prior to predict Hamiltonian matrices for the ground-state solution of DFT. The incorporation of SE(3)-invariant priors (GOE and TE) are novel and well-motivated by the symmetry properties of Hamiltonians.

3. The experiments are comprehensive and rigorous, comparing QHFlow with other leading regression-based approaches (e.g QHNet, WANet, SPHNet) across various datasets. The various experiments and metrics strengthen the validity of the findings.

**Weaknesses**
1. While QHFlow demonstrates strong performance on small to medium-sized systems (e.g., MD17, QH9), including out-of-distribution results within QH9, its scalability to larger and more complex molecules such as those in MD22 is untested. Evaluating larger molecules would help establish applicability to larger chemistry workloads and help strengthen the paper even further.

2. QHFlow primarily extends QHNet with the flow matching and symmetry-aware priors. Although, these proposed components lead to significant empirical improvements, the novelty of the approach is limited as it largely builds on existing methodologies and does not significantly advance methodology.

---

> ### Author Rebuttal · Authors · 2025-07-31
>
> Dear Reviewer upG2,
>
> Thank you for your thoughtful and encouraging review. We sincerely appreciate your positive feedback on the clarity of the manuscript, the novelty and motivation of our SE(3)-invariant priors, and the rigor of our experimental evaluations. Below, we address your comments in further detail.
>
> ---
>
> **W1, Q1: Lack of evaluation on larger molecules such as MD22.**
>
> We fully resonate with the reviewer's concern. However, this is infeasible at the moment since there are no public datasets with large molecule sizes to evaluate our method. Note that the MD22 dataset is not a quantum Hamiltonian dataset. We also note the existence of PubChemQH, which consists of molecules with 40–100 atoms, but the dataset is not public.
>
> Nevertheless, we strongly believe that our improvements will generalize to larger systems. The improvements may be larger since flow matching (and continuous-time generative models) has proven to be quite scalable for large systems. We note that our computational cost is similar to that of predictive models for each training step. For inference, the cost of multiple ODE integration steps is negligible compared to SCF iterations.
>
> ---
>
> **W2: The novelty of the approach is limited as it largely builds on existing methodologies and does not significantly advance methodology.**
>
> We resonate with this concern. Nevertheless, we would like to re-emphasize that our core contribution is not to methodological novelty. Instead, we propose to approach the Hamiltonian prediction problem with generative models to incorporate the underlying structure, which demonstrates strong empirical performance. We consider this idea to be novel, since the existing works mainly focused on developing new predictive architectures instead of generative models.
>
> ---
>
> **Q2: Is there any intuition why the TE prior generally performs better than the GOE prior?**
>
> Regrettably, this is more of an empirical finding rather than an insightful design. We hypothesize that the observed performance difference arises from the structural alignment between the TE prior and the Hamiltonian matrices. Specifically, the TE prior utilizes tensor expansion, which mirrors the structure and operations used to construct the Hamiltonian matrix in our model. Rather, GOE is constructed by the element-wise Gaussian distribution, which lacks this structural inductive bias.
>
> ---
>
> **Q3: There is generally a one-to-one mapping between structure and the ground-state Hamiltonian. Clarification on “distributional structure and uncertainty.**
>
> Thank you for pointing this out. We agree that the Hamiltonian is deterministic for a given structure. Our goal is not to model the uncertainty in data, but to better capture the predictive uncertainty arising from model approximation. In what follows, we provide a similar argument to Han et al. [1].
>
> We first clarify what we meant by distributional structure and uncertainty (in the prediction). Regression models typically minimize element-wise losses (e.g., MSE), implicitly modeling conditional independence between matrix elements: $p(\mathbf{H}|\mathcal{M}) = \prod_{i,j} p(H_{ij}|\mathcal{M})$, where $\mathbf{H}$ is the Hamiltonian, $H_{ij}$ is its $(i,j)$-th element, and $\mathcal{M}$ denotes the input molecule. In contrast, conditional generative models, such as diffusion or flow-based models, introduce a prior $z$ that globally conditions the output (i.e. $p(\mathbf{H}|\mathcal{M}) = \int_z p(z)\prod_{i,j} p(H_{ij}|\mathcal{M},z)dz$). This allows the model to better capture structural correlations among Hamiltonian elements. Even in deterministic tasks, this structure-aware generation leads to more coherent predictions, as evidenced in recent advances like AlphaFold3 [2], DiffCSP [3], image segmentation [4], and an object detection model [5].
>
> We further demonstrate how better capturing the uncertainty leads to improved performance. When the model is uncertain between two plausible Hamiltonians $\mathbf{H}_1$ and $\mathbf{H}_2$, a regression model often predicts their average $(\mathbf{H}_1 + \mathbf{H}_2)/2$, which may not correspond to a physically meaningful solution and results in high uncertainty. A generative model, however, can represent both modes explicitly, yielding sharper, more reliable outputs.
>
> Empirically, our flow-based model achieves consistently lower MAE compared to regression baselines as shown in Tables 1 and 2, supporting its advantage in modeling both the structural dependencies and predictive uncertainty of Hamiltonian prediction.
>
> Nevertheless, we fully agree that our earlier phrasing was ambiguous. We will revise the manuscript to clarify this point:
>
> >  (lines 38-39) These approaches rely on pointwise regression, which may lead to high predictive uncertainty and often struggle to capture the structural correlations present in the Hamiltonian matrix.
>
> ---
>
> **Suggestions and Typos**
>
> Thank you for your helpful suggestions. We will revise Section 5.4 to clearly indicate that the results shown in Table 3 correspond to QHFlow with the TE prior, as reported in Table 2.
>
> Specifically, we will update section 5.4 as follows:
> > (line 333-334) Table 3 compares the performance of GOE and TE priors, **where the TE prior results are identical to those of QHFlow reported in Table 2.** We found that the TE prior consistently yields lower errors than the GOE prior across all QH9 splits. This highlights the importance of designing task-aligned and symmetry-aware priors for accurate Hamiltonian prediction.
>
> We mark the text in bold to highlight its relevance to your concerns. They will be removed from the final manuscript.
>
> We also appreciate the correction regarding the molecular name in Table 1 and will correct it in the final version.
>
> ---
>
> We hope our responses have addressed your questions and concerns. Please let us know if there is anything we may have missed or if any points require further clarification.
>
> **References**
>
> [1] Han, Xizewen, Huangjie Zheng, and Mingyuan Zhou. "Card: Classification and regression diffusion models." NIPS 2022
>
> [2] Abramson, J., Adler, J., Dunger, J. et al. Accurate structure prediction of biomolecular interactions with AlphaFold 3. Nature 2024
>
> [3] Jiao, Rui, et al. "Crystal structure prediction by joint equivariant diffusion." NIPS 2023.
>
> [4] Chen, Shoufa, et al. "Diffusiondet: Diffusion model for object detection." ICCV 2023.
>
> [5] Baranchuk, Dmitry, et al. "Label-efficient semantic segmentation with diffusion models." ICLR 2022.

---

> ### Comment · Reviewer_upG2 · 2025-08-01
>
> Thank you for your detailed and thoughtful response to my comments, particularly the clarification regarding “distributional structure and uncertainty.”
>
> Regarding the large-scale evaluation, I was referring to the experimental setting in [1] in section 3.3, where the authors evaluated models on 500 randomly sampled conformations and applied the same SCF setup used in MD17 for the ground truth. However, since I did not cite this work in my original review, I do not expect you to incorporate this result within the discussion period though this could be a nice addition for the camera-ready as it would strengthen your argument of generalizing to larger systems.
>
> I have no further questions or concerns.
>
> [1] Zhang et al. “Self-Consistency Training for Density-Functional-Theory Hamiltonian Prediction.” ICML 2024

---

> ### Author Response · Authors · 2025-08-03
> **Response to Reviewer upG2**
>
> Thank you for your thoughtful comments and questions throughout the review process. Your feedback has helped us clarify and strengthen our work. We have reviewed the paper you mentioned and now have a clearer understanding of the context you referred to. We will incorporate the suggested evaluation in the camera-ready version, as we believe it will also be valuable for future follow-up studies.

---

### Official Review · Reviewer_NZgB · 2025-06-17

**Clarity:** 3
**Significance:** 2
**Originality:** 2
**Rating:** 5
**Confidence:** 4

**Summary:**

The authors introduce QHFlow, a flow matching method for predicting converged Kohn-Sham density functional theory (KS-DFT) Hamiltonians. The authors introduce two novel SO(3)-invariant base densities in the same vector space of KS-DFT Hamiltonians. This enables the authors to train a flow matching model conditioned on the molecular structure mapping from one of the base densities to the Dirac distribution of the converged Hamiltonian. In their experiments, the authors find that their QHFlow reduces MAE errors significantly compared to direct Hamiltonian predictors.

**Questions:**

* Figure 3 confused me as the caption and the appendix read that the SCF T Ratio is the ratio of converging a DFT calculation starting from QHFlow vs minao. Though, then a factor of 46 would mean that the QHFlow initialization takes 46 times as long to converge as minao? However, the text reads it's 13-15% faster? An explanation on how the figure should be read would be greatly appreciated.
* How sensitive is your approach to the choice of hyperparameters? Is there a good way to estimate these a priori?

**Ethical Concerns:**

["NO or VERY MINOR ethics concerns only"]

**Final Justification:**

I thank the reviewers for their thorough responses and increased my score accordingly.

**Limitations:**

No, the authors could discuss limitations such as scalability. It appears that the method does not scale to large systems, as the largest ones included here are from QM9.

**Quality:**

3

**Strengths And Weaknesses:**

Strengths:
* The paper is well written, easy to follow, and provides the necessary background for understanding their work.
* The authors propose an interesting application for flow matching-based regression.

Weaknesses:
* While the empirical performance motivates the work very well, the qualitative reasoning provided is weak. For instance, in line 39, it reads "[...] approaches [...] often fall short of capturing the full distributional structure and uncertainty inherent in the Hamiltonian." With the exception of degenerate states, there's neither a distribution of Hamiltonian nor an uncertainty for a given structure. As you correctly note yourself in l.169, where you describe it as a Dirac distribution.
* The technical novelty of this work is constrained to the two new base densities.
* Hyperparameters vary quite a bit between datasets, and suggestions such as the WA Loss provide inconsistent results.

---

> ### Author Rebuttal · Authors · 2025-07-31
>
> Dear Reviewer NZgB,
>
> Thank you for your thoughtful review. We appreciate your comments on the clarity of the writing with necessary background for understanding our work, and the proposed application of flow matching. Below, we address your points in detail.
>
> ---
>
> **W1: While the empirical performance motivates the work very well, the qualitative reasoning provided is weak. There is neither a distribution of Hamiltonian nor an uncertainty for a given structure.**
>
> Thank you for pointing this out. We agree that the Hamiltonian is deterministic for a given structure. Our goal is not to model the uncertainty in data, but to better capture the predictive uncertainty arising from model approximation. In what follows, we provide a similar argument to Han et al. [1].
>
> We first clarify what we meant by distributional structure and uncertainty (in the prediction). Regression models typically minimize element-wise losses (e.g., MSE), implicitly modeling conditional independence between matrix elements: $p(\mathbf{H}|\mathcal{M}) = \prod_{i,j} p(H_{ij}|\mathcal{M})$, where $\mathbf{H}$ is the Hamiltonian, $H_{ij}$ is its $(i,j)$-th element, and $\mathcal{M}$ denotes the input molecule. In contrast, conditional generative models, such as diffusion or flow-based models, introduce a prior $z$ that globally conditions the output (i.e. $p(\mathbf{H}|\mathcal{M}) = \int_z p(z)\prod_{i,j} p(H_{ij}|\mathcal{M},z)dz$). This allows the model to better capture structural correlations among Hamiltonian elements. Even in deterministic tasks, this structure-aware generation leads to more coherent predictions, as evidenced in recent advances like AlphaFold3 [2], DiffCSP [3], image segmentation [4], and an object detection model [5].
>
> We further demonstrate how better capturing the uncertainty leads to improved performance. When the model is uncertain between two plausible Hamiltonians $\mathbf{H}_1$ and $\mathbf{H}_2$, a regression model often predicts their average $(\mathbf{H}_1 + \mathbf{H}_2)/2$, which may not correspond to a physically meaningful solution and results in high uncertainty. A generative model, however, can represent both modes explicitly, yielding sharper, more reliable outputs.
>
> Empirically, our flow-based model achieves consistently lower MAE compared to regression baselines as shown in Tables 1 and 2, supporting its advantage in modeling both the structural dependencies and predictive uncertainty of Hamiltonian prediction.
>
> Nevertheless, we fully agree that our earlier phrasing was ambiguous. We will revise the manuscript to clarify this point:
>
> >  (lines 38-39) These approaches rely on pointwise regression, which may lead to high predictive uncertainty and often struggle to capture the structural correlations present in the Hamiltonian matrix.
>
> ---
>
> **W2: The technical novelty of this work is constrained to the two new base densities.**
>
> We resonate with this concern. Nevertheless, we would like to re-emphasize that our core contribution is not to methodological novelty. Instead, we propose to approach the Hamiltonian prediction problem with generative models to incorporate the underlying structure, which demonstrates strong empirical performance. We consider this idea to be novel, since the existing works mainly focused on developing new predictive architectures instead of generative models.
>
> ---
>
> **W3-1, Q2: Hyperparameters vary quite a bit between datasets. How sensitive is your approach to the choice of hyperparameters? Is there a good way to estimate these a priori?**
>
> We appreciate the reviewer’s concern and would like to clarify that our framework is generally robust to hyperparameter choices. Across seven out of eight datasets, we use nearly identical training hyperparameters, except for the minimum learning rate and the use of S-block embedding. The only exception was MD17–water, which contains only 500 training samples; For this case, we adopted a smaller learning rate to ensure stable optimization while keeping the other model hyperparameters fixed.
>
> As a general guideline, we recommend selecting the largest batch size based on available GPU memory, since increasing batch size or training steps consistently leads to improved performance. For other hyperparameters, we generally recommend following our setting. However, hyperparameter tuning is inevitable for achieving the best performance in machine learning. One can start by tuning the learning rates to further improve performance.
>
> ---
>
> **W3-2: WA Loss provides inconsistent results.**
>
> While we do not fully understand the reviewer’s concern, we believe it refers to the inconsistent improvements of QHFlow+WA-FT on Hamiltonian MAE in Table 2. For this case, we would like to clarify that this is, in fact, consistent with the motivation of WA-FT loss.
>
> We first note that the WA-FT loss was mainly designed to enhance energy prediction performance, which is consistently improved across datasets. When minimizing the weighted combination of the original QHFlow loss (for Hamiltonian prediction) and the WA-FT loss (for energy prediction), there exists a natural trade-off between the metrics. Our experiments show that one can find a "sweet spot" that yields high accuracies for both metrics, but the trade-off is inevitable.
>
> ---
>
> **Q1: Unclear explanation of Figure 3 (SCF accelerating performance)**
>
> Thank you for pointing this out. We will revise the explanation of Figure 3 (both the caption and the main text) to improve clarity.
>
> Specifically, we will update the caption as follows:
> > (Figure 3 caption) The SCF Iter Ratio reflects the relative improvement in the number of SCF iterations compared to conventional DFT. All time-related values (Inf T Ratio, SCF T Ratio, Total T Ratio) are expressed as percentages relative to the total runtime of conventional DFT initialized with minao, which is set to 100%. Lower values indicate faster convergence. For example, an “Total T Ratio” of 46% means that QHFlow converges in 46% of the time required by standard DFT, including the model inference time.
>
> In the main text, we will revise Section 5.3 as follows:
> > (line 321-323) As shown in Figure 3, QHFlow significantly accelerates SCF convergence. For instance, on QH9-stable (id), QHFlow achieves convergence in just 46% of the runtime compared to conventional DFT, corresponding to a 54% relative speedup. Compared to prior ML-based initializations such as QHNet (53%), QHFlow yields an additional 7%p reduction. These results demonstrate the practical utility of QHFlow for accelerating real-world DFT simulations.
>
> While we originally reported relative ratios (i.e., 13\% $\approx 1 - \frac{0.46}{0.53}$), we agree that we should present the improvement more clearly, such as expressing it as a percentage-point difference (e.g., a 7%p gain from 53% to 46%).
>
> ---
>
> **Limitations: Lack of discussion on limitations in terms of scalability**
>
> Thank you for pointing this out! We originally included the discussion in Appendix F.4, but now realize its low visibility despite the importance. To alleviate your concern, we will move the limitation to the main text and revise the contents.
>
> To be specific, we will include the following text:
> > We did not fully verify the generalizability of our approach, e.g., extension to new architectures (WANet, SPHNet) or datasets (PubChemQH), due to the lack of resources, public codes, and public datasets. Our flow-based formulation introduces computational overhead for solving ODEs. In the future, one could consider removing the overhead with the one-step generative models [6]. Our experiments are limited to gas-phase molecules with up to 29 atoms and two commonly used XC functionals (PBE and B3LYP). The algorithms are left to be verified for more general settings, such as periodic solids or higher-accuracy functionals (e.g., hybrid or double-hybrid). **We also acknowledge that our datasets consist of relatively small molecules. One could further test the scalability of our approach once new datasets on larger systems are available.** We also note that our work does not give a theoretical explanation of how Hamiltonian prediction error translates to SCF acceleration. Developing a theoretically grounded algorithm specifically for SCF acceleration would be an interesting future research.
>
> We mark the text in bold to highlight its relevance to your concerns. They will be removed from the final manuscript.
>
> ---
>
> We hope our responses have addressed your questions and concerns. Please let us know if there is anything we may have missed or if any points require further clarification.
>
> **References**
>
> [1] Han, Xizewen, Huangjie Zheng, and Mingyuan Zhou. "Card: Classification and regression diffusion models." NIPS 2022
>
> [2] Abramson, J., Adler, J., Dunger, J. et al. Accurate structure prediction of biomolecular interactions with AlphaFold 3. Nature 2024
>
> [3] Jiao, Rui, et al. "Crystal structure prediction by joint equivariant diffusion." NIPS 2023.
>
> [4] Chen, Shoufa, et al. "Diffusiondet: Diffusion model for object detection." ICCV 2023.
>
> [5] Baranchuk, Dmitry, et al. "Label-efficient semantic segmentation with diffusion models." ICLR 2022.
>
> [6] Kornilov, Nikita, et al. "Optimal flow matching: Learning straight trajectories in just one step." 2024 NIPS.

---

### Official Review · Reviewer_YiLn · 2025-07-03

**Clarity:** 3
**Significance:** 4
**Originality:** 3
**Rating:** 5
**Confidence:** 4

**Summary:**

DFT is an approximation to solve the Schroedingers Equation. It has received nobel prizes. DFT is computationally demanding and takes hours of compute for a single molecule. In the past, starting with Snyder et al PRL 2012 ML approaches have been used to estimate DFT densities. Other work followed. Also a direct prediction of the Hamiltonian has been proposed (Schütt et al 2019). In the meantime a number of approaches have been conceived. Also some that attempt to directly improve and accelerate the DFT loop.

The present ms proposes to use higher-order flow matching for DFT level Hamiltonian prediction in an equivariant fashion and presents impressive improvements over SOTA. Also SCF iterations in the DFT loop can be accelerated using the proposed approach.

**Questions:**

One area that could benefit from further clarification is the chemical meaning of the observed progress. While the results are promising, it would strengthen the paper to elaborate more explicitly on what the improvements represent in terms of underlying chemical principles or phenomena. Providing a clearer interpretation from a chemical standpoint would help bridge the gap between the computational findings and their practical or theoretical implications in chemistry.

Additionally, the paper would be improved by a more detailed discussion of the limitations of the proposed method. While the results are strong, every approach has its constraints, and a candid evaluation of where the method may fall short—whether in terms of scalability, generalizability, computational cost, or domain-specific applicability—would offer a more balanced perspective. Such a discussion would not only increase the transparency of the work but also guide future research directions.

**Ethical Concerns:**

["NO or VERY MINOR ethics concerns only"]

**Final Justification:**

I stay with my assessment in the light of the rebuttal and discussion. I would accept the ms.

**Limitations:**

yes

**Quality:**

3

**Strengths And Weaknesses:**

strength

The paper is clearly written, with well-structured sections and precise language that make it easy to follow the authors' arguments and methodology. The clarity of presentation contributes significantly to the overall readability and accessibility of the work.

The experimental section is impressive. The authors conduct a thorough and well-designed set of experiments that convincingly support their claims. The results are clearly presented, and the comparison with relevant baselines demonstrates the effectiveness of the proposed approach.

The paper is also well-situated within the existing body of literature. The authors provide a comprehensive and thoughtful review of related work, and they effectively position their contribution in relation to previous studies. This contextual grounding enhances the reader’s understanding of the paper’s novelty and relevance.


weaknesses
- it remains unclear to reviewer from the ms what the differences in the tables mean chemically, i.e. are the improvements insightful or chemically meaningless?
(To give context to this question: many newer ML approaches in MLFFs provide improvements that are chemically meaningless and yield unstable MDs)
- I haven't found limits discussed, please discuss them carefully

---

> ### Author Rebuttal · Authors · 2025-07-30
>
> Dear Reviewer YiLn,
>
> Thank you for your thoughtful and encouraging review. We appreciate your recognition of the clarity of our presentation, the strength of our experiments, and the relevance of our contribution within the existing literature. Below, we address your comments in detail.
>
> ---
>
> **W1, Q1: Are the improvements insightful or chemically meaningless?**
>
> Our improvements are chemically meaningful since they (1) reliably accelerate DFT calculations and (2) maintain chemical accuracy even for out-of-distributions.
>
> First, since QHFlow accelerates DFT (as shown in Fig. 3), it reduces the overall computational cost in cases where DFT was previously required. Even for out-of-distribution data, the additional SCF iterations ensure that the solver reliably achieves DFT-level accuracy. This is particularly impactful for chemical simulations, such as *ab initio* molecular dynamics (AIMD), which heavily rely on DFT calculations and often encounter out-of-distribution inputs.
>
> Furthermore, even without additional SCF iterations, QHFlow achieves the level of chemical accuracy (error < 1 kcal/mol ≈ 1,600 μEₕ [1,2]) for energy-related properties such as HOMO, LUMO, and the HOMO–LUMO gap, where baseline models fall short. Notably, as shown in Table 2, this level of accuracy is maintained even on unseen molecular configurations from molecular dynamics trajectories, specifically the QH9-dynamic (300k-mol) set.
>
> ---
>
> **W2, Q2: Lack of discussion on limitations in terms of scalability, generalizability, computational cost, or domain-specific applicability.**
>
> Thank you for the suggestion! We originally included the discussion in Appendix F.4, but now realize that its visibility is low despite its importance. To alleviate your concern, we will move the limitation to the main text and add more content to incorporate your comments.
>
> To be specific, we will include the following text:
> > **(Generalizability)** We did not fully verify the generalizability of our approach, e.g., extension to new architectures (WANet, SPHNet) or datasets (PubChemQH), due to the lack of resources, public codes, and public datasets. **(Computational cost)** Our flow-based formulation introduces computational overhead for solving ODEs. In the future, one could consider removing the overhead with the one-step generative models [3]. **(Domain-specific applicability)** Our experiments are limited to gas-phase molecules with up to 29 atoms and two commonly used XC functionals (PBE and B3LYP). The algorithms are left to be verified for more general settings, such as periodic solids or higher-accuracy functionals (e.g., hybrid or double-hybrid). **(Scalability)** We also acknowledge that our datasets consist of relatively small molecules. One could further test the scalability of our approach once new datasets on larger systems are available. We also note that our work does not give a theoretical explanation of how Hamiltonian prediction error translates to SCF acceleration. Developing a theoretically grounded algorithm specifically for SCF acceleration would be an interesting future research.
>
> We mark the keywords, e.g., generalizability and computational cost, as bold to highlight their relevance to your concerns. They will be removed from the final manuscript.
>
> ---
>
> We hope our responses have addressed your questions and concerns. Please let us know if there is anything we may have missed or if any points require further clarification.
>
> **Reference**
>
> [1] Pople, John A. "Nobel lecture: Quantum chemical models." Reviews of Modern Physics 71.5 (1999): 1267.
>
> [2] Kuang, Yang, Yedan Shen, and Guanghui Hu. "Towards chemical accuracy using a multi-mesh adaptive finite element method in all-electron density functional theory." Journal of Computational Physics 518 (2024): 113312.
>
> [3] Kornilov, Nikita, et al. "Optimal flow matching: Learning straight trajectories in just one step." 2024 NIPS.

---

> > ### Comment · Reviewer_YiLn · 2025-08-07
> >
> > I thank the authors for their answers and stay with my good assessment of the work.

---

> ### Author Response · Authors · 2025-08-09
>
> Thank you for your kind support of our work. We truly appreciate your constructive comments throughout the review process, which have helped us improve the overall quality of our work.

---

### Comment · Area_Chair_Ve7D · 2025-08-08

Thank you, reviewers and authors, for your active and productive engagement during the Author+Reviewer discussion period. Based on the comments so far, it appears there are no outstanding questions requiring further author response. If that’s not the case, please add any final comments as soon as possible before the end of today (August 8 AoE), ahead of the AC+Reviewer discussion phase.

I note that one reviewer has not yet provided a full rebuttal response, but the remaining discussions appear to be resolved.

---

### Note · Authors · 2025-08-13

Dear Reviewers and Area Chair,

Thank you for the constructive and encouraging discussions. We are pleased that all reviewers reached an agreement to accept our paper prior to the rebuttal period (5/4/5/5). We appreciate the recognition of our work for its novelty in applying flow matching to Hamiltonian prediction in a symmetry-aware manner (all), the clarity of our writing (all), the inclusion of necessary background (YiLN, NZgB), and the significant performance gains over baselines (all).

We are grateful for the collaborative and productive exchanges. All raised concerns were resolved through discussions with the engaged reviewers, and their suggestions have been incorporated. The corresponding revisions will be reflected in the future manuscript, as detailed in the individual reviewer threads.

Below is a brief summary of the discussion outcomes:
- **Scalability to larger molecules.** We addressed the concerns about scalability through discussion, and we will further strengthen this in the final version with additional experiments on larger systems (e.g., MD22).
- **Ambiguity regarding Hamiltonian uncertainty.** We clarified this by distinguishing between data uncertainty and predictive uncertainty, and we will revise the manuscript to explicitly note that we refer to predictive uncertainty.
- **Lack of limitations.** We originally included the limitation section in Appendix F.4. We will move this section to the main text and expand it to address scalability, generalizability, computational cost, domain-specific applicability, and the weak correlation between Hamiltonian error and SCF acceleration.

We sincerely appreciate the reviewers’ constructive feedback, which has strengthened our work, and we will reflect all changes discussed in the final manuscript.

Best regards,

Authors

---

### Decision · Program_Chairs · 2025-09-17

**Decision:**

Accept (spotlight)

**Comment:**

## **Summary of the Paper**
This paper proposes **QHFlow**, an $\mathrm{SE}(3)$-equivariant generative model for predicting Kohn–Sham Hamiltonians in a localized atomic orbital basis. Unlike prior deterministic regression approaches (e.g., QHNet), QHFlow frames prediction as a *flow matching* problem, mapping from a symmetry-aware prior (Gaussian Orthogonal Ensemble or Tensor Expansion) to the target Hamiltonian. The method also incorporates architectural changes such as attention-based message passing, channel mixing, and use of the overlap matrix $S$ as an additional input. The model is evaluated on MD17 and QH9 datasets, showing improved Hamiltonian MAE, derived property accuracy, and reduced SCF wall-time.

## **Strengths**
- Addresses a relevant and active problem in quantum chemistry with a novel generative framing.
- Consistent, clear empirical improvements over strong baselines (QHNet, WANet, SPHNet) across multiple datasets.
- Incorporates physically meaningful priors and evaluates on metrics directly relevant to downstream SCF convergence.
- Well-written and accessible, with relevant background context.

## **Weaknesses**
- Main novelty (flow matching) is introduced alongside several other changes — notably the use of structured priors so the model is not starting from scratch, inclusion of $S$, and architectural updates— without sufficient ablations to attribute gains specifically to the generative formulation.
- Scalability to larger systems not demonstrated (currently being addressed by authors with additional experiments on MD22).
- Minor presentation issues (initial ambiguity in “uncertainty” wording, figure clarity) were largely addressed in rebuttal.

## **Reviewer Consensus and Discussion**
Reviewers agree the performance gains are convincing and the generative approach is interesting, but several noted that attribution is unclear. In particular, the introduction of $S$ as an input and the use of structured priors are significant changes from baselines and may account for part of the improvement. While the rebuttal included some ablations, I recommend two **relatively light additional experiments** that could still be conducted during the camera-ready phase to clarify attribution and strengthen the contribution:
1. QHFlow **without** $S$, to isolate the effect of the overlap matrix.
2. A QHNet variant trained to predict $\Delta H$ from the same prior initialization (e.g., TE prior), to separate the effect of the structured prior from the flow mechanism.

**Note:** These ablations are my own suggestions as AC to address reviewer and community concerns—they were not explicitly requested by the reviewers. The NeurIPS 2025 camera-ready deadline is **October 23, 2025 AoE**, which may still allow time to run these tests before finalizing the manuscript.

## **Recommendation**
I recommend **Accept (Spotlight)**. This is the highest-scoring paper in my AC batch. The contribution is timely, broadly relevant, and demonstrates strong empirical gains over strong baselines, warranting wider visibility. While some attribution questions remain, they do not diminish the significance or potential impact of the work. Including the suggested ablations before the camera-ready deadline could make the case for the generative formulation even stronger.